# Polo kinase recruitment via the constitutive centromere-associated network at the kinetochore elevates centromeric RNA

**Guðjón Ólafsson**[ID], **Peter H. Thorpe**[ID]*

School of Biological and Chemical Sciences, Queen Mary, University of London, London, United Kingdom

* p.thorpe@qmul.ac.uk

## Abstract

The kinetochore, a multi-protein complex assembled on centromeres, is essential to segregate chromosomes during cell division. Deficiencies in kinetochore function can lead to chromosomal instability and aneuploidy—a hallmark of cancer cells. Kinetochore function is controlled by recruitment of regulatory proteins, many of which have been documented, however their function often remains uncharacterized and many are yet to be identified. To identify candidates of kinetochore regulation we used a proteome-wide protein association strategy in budding yeast and detected many proteins that are involved in post-translational modifications such as kinases, phosphatases and histone modifiers. We focused on the Polo-like kinase, Cdc5, and interrogated which cellular components were sensitive to constitutive Cdc5 localization. The kinetochore is particularly sensitive to constitutive Cdc5 kinase activity. Targeting Cdc5 to different kinetochore subcomplexes produced diverse phenotypes, consistent with multiple distinct functions at the kinetochore. We show that targeting Cdc5 to the inner kinetochore, the constitutive centromere-associated network (CCAN), increases the levels of centromeric RNA via an *SPT4* dependent mechanism.

## Author summary

During cell division, replicated chromosomes must be equally divided between the two daughter cells. This is achieved in part by a multi-protein structure called the kinetochore. Errors during chromosome segregation can lead to inheritance of abnormal chromosome number. This situation is referred to as aneuploidy—a hallmark of cancer cells. Kinetochore function is tightly regulated by recruitment of regulatory proteins under certain conditions and at specific times during the cell cycle. Kinetochore regulators are often mis-regulated in many types of cancers. The major cell-cycle regulator, Polo-like kinase (Plk1), is one such regulator. Plk1 is overexpressed in many cancers and is thus considered an important target in cancer therapeutics. To understand its function in kinetochore regulation, we manipulated the Plk1 homolog in budding yeast, Cdc5, in both space and time. Our data suggest that Cdc5 has multiple different functions at the yeast kinetochore in agreement with studies in human cells. Furthermore, we show that constitutive Cdc5 localization at the inner kinetochore can disrupt the function of the whole kinetochore,

**Citation:** Ólafsson G, Thorpe PH (2020) Polo kinase recruitment via the constitutive centromere-associated network at the kinetochore elevates centromeric RNA. PLoS Genet 16(8): e1008990. https://doi.org/10.1371/journal.pgen.1008990

**Data Availability Statement:** All relevant data are within the manuscript and its Supporting Information files.

**Funding:** G.O. and P.H.T. received funding from the Francis Crick Institute (FC001003; www.crick.

ac.uk). The funders had no role in study design, data collection and analysis, decision to publish, or preparation of the manuscript.

**Competing interests:** The authors have declared that no competing interests exist.

perturb mitotic progression and increase the levels of centromeric RNA. These results have implications for human disease, since mis-regulation of centromeric transcription has been observed in stressed, aging and cancerous cells.

## Introduction

The kinetochore anchors chromosomes via their centromeres to the spindle microtubules during cell division. The budding yeast centromere is defined by a specific ~125 base-pair DNA sequence comprised of three domains termed centromere DNA elements I, II and III (CDEI-III) which interact with Cbf1, Cse4$^{CENP-A}$ nucleosome and the budding-yeast-specific CBF3 complex, respectively (reviewed in [1]). These inner kinetochore proteins interact with the 16-subunit Constitutive-Centromere Associated Network (CCAN) [2,3]. The CCAN binds to outer kinetochore complexes MIND$^{MIS12}$ and NDC80, which together with the yeast-specific DAM1/DASH complex provides a binding interface for a single microtubule emanating from the spindle pole body (SPB; the yeast centrosome) (reviewed in [4]). The yeast kinetochore is bound to microtubules throughout the cell cycle, except for a brief moment in early S-phase when the centromeric DNA undergoes replication [5,6]. The outer kinetochore also contains the SPC105$^{KNL1}$ complex which is the key regulatory hub to signal the spindle assembly checkpoint (SAC) to delay anaphase onset when kinetochores are not or incorrectly attached to microtubules. Together the SPC105$^{KNL1}$, MIND$^{MIS12}$ and NDC80 complexes are referred to as the KMN network.

Many kinetochore proteins are regulated by post-translational modifications such as phosphorylation, ubiquitylation, methylation and acetylation. For instance, to prevent chromosome missegregation Ipl1$^{Aurora\ B}$ kinase destabilizes incorrect microtubule attachments by phosphorylating subunits at the NDC80 and DAM1/DASH complexes. This creates unattached kinetochores which leads to the recruitment of Mps1 kinase to phosphorylate Spc105$^{Knl1}$ to activate the SAC signaling cascade that ultimately inhibits Cdc20, the activator of the Anaphase Promoting Complex or Cyclosome (APC/C) (reviewed in [7–10]). When proper attachment to the mitotic spindle has been achieved the SAC is silenced by phosphatase recruitment (PP1 and PP2A), to counteract Ipl1$^{Aurora\ B}$ and Mps1 kinase activity and stabilize correct microtubule attachments [11–17]. However, the function of many other modifications at the kinetochore are still uncharacterized and/or the enzymes in question remain unknown. A method to systematically associate any protein of interest with thousands of query proteins in live cells and assess the effects of such binary interactions on growth allows us to identify Synthetic Physical Interactions (SPIs) [18–20]. SPIs have been used to identify kinetochore regulators [18], to investigate SPB duplication [21] and to dissect the SAC [22].

Our previous proteome-wide kinetochore screens identified many candidates of kinetochore regulation–both uncharacterized and characterized–including the Polo-like kinase, Cdc5 [18]. Polo-like kinases (hereafter referred to as Polo kinase) are highly conserved in eukaryotes and tightly spatiotemporally regulated with multiple roles in cell-cycle progression (reviewed in [23]). Polo kinase is characterized by a N-terminal catalytic kinase domain that is flexibly linked to a binding domain called Polo-box domain (PBD) whose key function is to target the kinase domain to its substrates by binding preferentially phosphosites that have previously been phosphorylated by another kinase, most commonly by cyclin-dependent kinase (CDK) or Polo kinase itself (reviewed in [24]). Budding yeast Polo kinase, Cdc5, is known to interact with kinetochore proteins [25] and localize at centromeres in mitosis [26,27]. In our prior work we found that forced Cdc5 recruitment to the MIND complex resulted in a growth

defect and chromosomal instability [18]. More recently, we showed that Cdc5 phosphorylates Cse4 in a cell-cycle regulated manner [27]. The kinetochore function of Polo kinase in budding yeast remains elusive since Polo kinase mutations have pleiotropic effects on cells. However, fusion proteins have been effective tools to query Polo kinase function in both yeast and human cells [28–31].

In this study we focused on Cdc5. First, we compared the forced recruitment of Cdc5 with other kinases and candidates of kinetochore regulation by probing the entire kinetochore using the SPI system. We identified many proteins, including Cdc5, that perturb kinetochore function, some of which have known kinetochore roles whereas others have no known kinetochore function. Second, we performed a proteome-wide SPI screen with Cdc5 and forcibly recruited it to every GFP-tagged protein. We found that Cdc5 SPIs were enriched for kinetochore, RNA polymerase II transcription factor complex (RNAPII) and SAGA histone acetyltransferase complex proteins. Additionally, we found that forcing Cdc5 to several distinct kinetochore subcomplexes produced a growth phenotype in a kinase-dependent but SAC-independent manner. We also explored the forced interactions in more detail using fluorescence microscopy and discovered that depending on when and where at the kinetochore Cdc5 was recruited it produced different phenotypes. These findings support the notion that Polo kinase has a number of distinct roles at the kinetochore [30]. We investigated the forced association of Cdc5 with the CCAN subunit Ame1$^{CENP-U}$ in more detail and found that it caused declustering of the kinetochore and impaired metaphase to anaphase transition. Furthermore, we performed a genetic suppressor screen to detect the genetic dependencies of the growth phenotype. Using synthetic biology and genetics approaches, we found that Cdc5 recruitment specifically to the CCAN can increase centromeric (CEN) RNA, via a mechanism dependent upon the Spt4/5 complex, which is involved in transcription elongation and mRNA processing. This observation provides a potential mechanism to explain the cell cycle regulation of centromeric transcription or CEN RNA levels in budding yeast.

## Results

### Identifying kinetochore regulators using Synthetic Physical Interactions

Synthetic Physical Interactions (SPIs) have been successful in identifying proteins that regulate the kinetochore [18,20]. The SPI system (Fig 1A, bottom panel, and S1A Fig), which utilizes the GFP-binding protein (GBP; also known as a nanobody [32]) and the genome-wide GFP collection [33], allows us to artificially associate two proteins in live cells, which for most pairs does not affect cell growth [20]. However, for specific complexes within the cell, such as the kinetochore, the recruitment of regulator proteins is carefully controlled such that they act at a specific time during the cell cycle or under specific circumstances. For example, if kinetochores dissociate from the spindle microtubules, proteins are recruited to the kinetochore to activate the spindle assembly checkpoint. The rationale for using SPIs is that this spatiotemporal control can be disrupted with constitutive protein associations. Thus, a regulator is locked in place, regardless of cell cycle stage or circumstances. Since kinetochore function is essential, such mis-regulation is likely to result in a growth defect. Consistent with this notion, artificial associations of characterized regulators to the kinetochore perturbs growth [13,18,22,34,35].

To comprehensively identify kinetochore regulators, we extended our analysis across the whole kinetochore, since prior SPI screens were focused on outer kinetochore proteins at the KMN network and the DAM1/DASH complex (Mtw1$^{MIS12}$, Nuf2$^{NUF2}$ and Dad2). We performed SPI screens with two components of the CCAN, Ctf19$^{CENP-P}$ and Cnn1$^{CENP-T}$ (Fig 1A). Collectively, these data identify 203 GFP-tagged proteins that produced SPIs in at least one of the five proteome-wide kinetochore SPI screens (S1 Data). We mapped the kinetochore

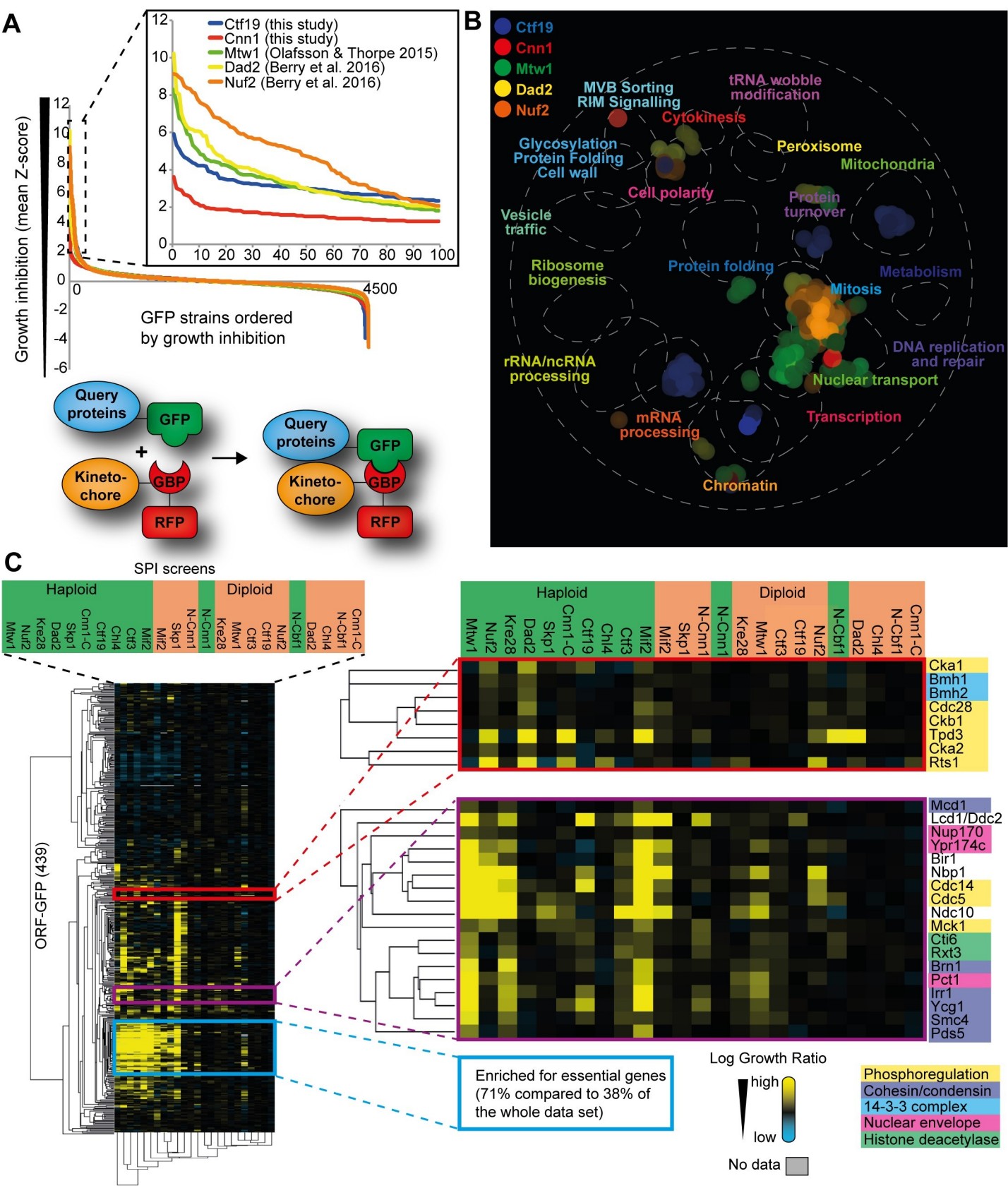

**Fig 1. Identification of kinetochore regulators using Synthetic Physical Interactions.** (A) Data from five proteome-wide kinetochore SPI screens are plotted in order of growth inhibition. Three SPI screens were performed previously (Mtw1, Nuf2 and Dad2) and two were performed in this study (Ctf19 and Cnn1). Growth inhibition caused by forced binary protein interactions are indicated on the y-axis is a mean Z-score (see Materials and Methods for more details). We took a Z-score of $\geq 2$ as a cutoff for a significant growth defect. The Z-score $\geq 2$ corresponds to a two-fold or greater difference in colony sizes. Most forced interactions do not cause a growth defect (Z-score $\approx 0$) whereas SPI screens such as Nuf2 and Dad2 have many SPIs ($>$100 strains with Z-score $\geq 2$). The Ctf19 and Cnn1 SPI data are listed in S1 Data. (B) The data from the five proteome-wide kinetochore SPI screens were mapped onto the global similarity network using spatial analysis of functional enrichment (SAFE) on thecellmap.org, which shows that the kinetochore SPI data is enriched for specific biological processes or cellular compartments; mainly mitosis, nuclear transport, mRNA processing and chromatin. (C) SPI data from 12 screens (in both haploid and heterozygous diploid strains) for 439 GFP-tagged strains were analyzed using Cluster software and visualized using Java TreeView. The strength of the growth inhibition (log growth ratio) is shown using a yellow-blue color scale where yellow is a strong growth defect compared to controls, black indicates no effect, and blue indicates growth enhancement. The clustering analysis distinctly clusters together the haploid SPI screens to the left (indicated in green) and diploid screens to the right (indicated in orange), with the exception of GBP-Cnn1 and GBP-Cbf1 screens. Furthermore, the cluster analysis clusters together GFP strains that are similarly affected by the GBP-tagged kinetochore proteins. Three distinct clusters are highlighted in the inset (see Materials and Methods and text for details).

SPI data onto the global genetic similarity network [36] using the online tool thecellmap.org [37] and we used SAFE analysis (Spatial Analysis of Functional Enrichment) to identify genetic network regions that are significantly enriched [38] (Fig 1B). Next, to both confirm and classify these 203 proteins in more detail, we assembled a collection of GFP strains including the 203 strains identified in the proteome-wide SPI screens and an additional 236 GFP strains that we bootstrapped in as either likely regulators, e.g. protein modifiers such as kinases, or as proteins involved in kinetochore function; in total 439 GFP-proteins (S2 Data). We generated six additional kinetochore proteins fused to GBP: Skp1, Cbf1CENP-B, Ctf3CENP-I, Chl4CENP-N, Mif2CENP-C and Kre28ZWINT1, while also retesting Mtw1, Nuf2, Dad2, Ctf19 and Cnn1 (Cnn1-GBP and GBP-Cnn1). We screened these 12 GBP-fusions, which we chose to represent members of every kinetochore subcomplex, against the collection of 439 GFP strains. We screened both haploid cells (which contain a single allele of each GFP-tagged gene) and heterozygous diploid cells (where the GFP allele is complemented with an untagged allele) to minimize possible effects of perturbing an essential function of certain GFP-tagged proteins.

This more refined approach measured the growth effects of over 10,000 binary interactions (S2 Data). Hierarchical cluster analysis of the data reveals that several groups of GFP-tagged strains produce similar growth defects (SPIs) when associated with different kinetochore components (Fig 1B). Using this approach, we can ask whether a kinase, such as Cdc5, would cluster with other kinases such as CDK or separately. First, we note that most haploid screens clustered separately from the diploid screens. This is caused by a group of GFP proteins (blue box highlighted in Fig 1C), which are sensitive to most GBP-fusions, as described previously [20]. Focusing on phosphoregulation, we find that another cluster group (red group in Fig 1C) includes kinases such as casein kinase 2 (CK2; Cka1, Cka2 and Ckb1) and Cdc28CDK1, and PP2A phosphatase subunits (Tpd3 and Rts1), and both components of the budding yeast 14-3-3 complex (Bmh1 and Bmh2). A third cluster group (purple box group in Fig 1C) which includes Cdc5, also contains the mitotic phosphatase Cdc14, Mck1GSK3 kinase and a Mec1ATR kinase recruitment factor Lcd1ATRIP (also known as Ddc2), together with members of the cohesin and condensin complexes. This group also consists of the SPB component Nbp1 and its uncharacterized paralog Ypr174c, both of which interact with Cdc5 [25,39]. Furthermore, this third cluster group includes two components of the Rpd3L histone deacetylase complex (HDAC), Rxt3 and Cti6. Although the second and the third cluster groups (Fig 1C, red and purple groups) both include kinases and phosphatases, their SPI profiles are quite distinct; For example in the second red cluster, CK2 gives the strongest SPI phenotype with mostly outer kinetochore proteins such as Nuf2 and Dad2 and overlaps with PP2A. In contrast, in the third purple cluster, Cdc5 produces the strongest SPI phenotype with Mtw1, Nuf2, Kre28 and Mif2, and overlaps with Cdc14 phosphatase. This suggests that, based upon our SPI data, the forced Cdc5 recruitment to different kinetochore subcomplexes has distinct characteristics compared

to kinases such as CDK and CK2 but also compared to other kinases that we have tested but are not highlighted in this figure (see S1C and S1D Fig for comparison with other kinases and regulators such as Dbf4-dependent kinase DDK and Ipl1$^{Aurora\ B}$).

The overlap between the haploid and diploid kinetochore SPI screens produces a list of 119 GFP strains in total that were identified as SPIs with at least one of the 12 GBP-tagged kinetochore proteins (S1A Fig). In this conservative estimate of kinetochore SPIs, any haploid SPIs that failed to give a growth phenotype in diploid cells were excluded (S1C and S1D Fig). Using this list of 119 kinetochore SPIs the outer and inner kinetochore SPIs (72 and 88 SPIs, respectively) produced an overlap of 41 SPIs (~35%) (S1B Fig). In addition to post-translational modifiers, such as kinases and phosphatases, we were surprised to find many proteins involved in transcription regulation, chromatin remodeling and RNA processing in our kinetochore SPI screens.

Taken together the kinetochore SPI data highlight three key points. First, that the effect of recruiting candidate regulators is specific for the individual subcomplexes within the kinetochore (S1C and S1D Fig). For instance, most of the HDAC proteins were specifically identified in the Mtw1 and Ctf3 SPI screens. Second, that both kinases and phosphatases cause kinetochore SPI phenotypes, suggesting that constitutive phosphorylation or dephosphorylation of kinetochore subcomplexes is detrimental for cell growth. Third, the mitotic kinase, Cdc5, stands out as producing phenotypes across the kinetochore—in particular with Mif2, Mtw1 and Nuf2—consistent with multiple functions at distinct kinetochore subcomplexes. Furthermore, the Cdc5 SPI phenotype is most similar to that of the key cell-cycle phosphatase Cdc14 (Fig 1C, purple group). Therefore, we wanted to determine how a misregulated Cdc5 activity at different kinetochore subcomplexes is perturbing kinetochore function.

## Constitutive Cdc5 kinase activity at different kinetochore subcomplexes disrupts cell growth

We chose to further investigate Cdc5 for multiple reasons. First, we previously identified Cse4 as a substrate of Cdc5 [27]. Second, Cdc5 has been shown to interact with both Cse4 and Ndc80 [25] and phosphorylate microtubule-associated proteins (MAPs) such as Stu2 and Slk19 [40] and the SAC component Mad3 [41]. Third, phosphorylation of Cse4 by both Cdc5 and Ipl1$^{Aurora\ B}$ kinase is important for faithful chromosome segregation [42]. Fourth, Cdc5 responds to CDK phosphorylation, making it a good candidate for facilitating or amplifying cell cycle signals at the kinetochore. Fifth, phosphorylation of other kinetochore proteins by Cdc5 remains largely unexplored in budding yeast. Finally, in our SPI screen we identified Bbp1 as a SPI with most kinetochore proteins (9/12; S2 Data) and both Nbp1 (a binding partner of Bbp1) and its paralog, Ypr174c, as kinetochore SPIs with a similar SPI phenotype to Cdc5 (Fig 1B). Ypr174c, which is proposed to be an anchoring scaffold for Cdc5 [43], Bbp1 and Nbp1 all interact with Cdc5 [25,39]. Nbp1 is a component of the Mps2-Bbp1 complex which associates with the SPB and is important for mitotic function of Cdc5 [29]. Interestingly, the *S. pombe* Bbp1 homolog, Sos7, is a kinetochore component [44]. In *S. cerevisiae*, Bbp1 interacts with the CCAN proteins, Nkp2 and Mcm22, and Nbp1 and Ypr174c interact with Ame1 and Ctf19, respectively [45,46]. Ame1 and Ctf19 are both components of the COMA kinetochore subcomplex which is part of the CCAN. Our kinetochore SPI data suggest that Cdc5 may function at the inner and the outer kinetochore in agreement with findings in both yeast and human cells, which together suggest it may have multiple functions across the kinetochore [25,30].

We subsequently asked which proteins in the whole proteome are most sensitive to Cdc5 localization by fusing Cdc5 to GBP and transferring it to the entire GFP collection. This

proteome-wide approach allows us to examine the growth defects caused by Cdc5 interaction with the kinetochore in the context of all other forced interactions throughout the cell. We used a plasmid encoding Cdc5-GBP and control plasmids containing untagged Cdc5 to control for ectopic expression of Cdc5 as well as a plasmid containing a kinase-deficient version of Cdc5 (cdc5-T242A; [47]) to investigate the catalytic dependency of the SPI phenotype. We note that some GFP-tagged proteins were sensitive to the kinase-dead Cdc5, likely because the Polo-box domain (PBD) may associate with Cdc5 substrates, hence to minimize any possible interference we removed the PBD phospho-binding activity by mutating three residues in the PBD [25,48,49] (cdc5-kd-PBD*-GBP; Fig 2A and 2B). We combined each of these Cdc5 plasmids separately with the members of the GFP collection and scored the resulting strains for growth as previously described. We find that approximately 100 GFP strains show growth defects when comparing the active kinase with any of the controls (Fig 2C and S3 Data). What was striking from these data was the number of kinetochore proteins represented in this set of Cdc5 SPIs (18 of the 33 proteins annotated as "kinetochore component", were in the top 62 SPIs; GO enrichment p value = $3 \times 10^{-15}$; Table 1). We demonstrated that Cdc5 was constitutively recruited to many different cellular locations, including the kinetochore using fluorescence microscopy (Fig 2D and S2A and S2B Fig).

In addition to the kinetochore, gene ontology (GO) enrichment analysis of the Cdc5 SPI data, also identifies components of the histone acetyltransferase SAGA complex and RNAPII transcription factor complex as sensitive to constitutive Cdc5 association (Table 1). It was surprising to us how many kinetochore proteins were detected in the proteome-wide SPI screen, we thus decided to rescreen Cdc5 to ask where within the kinetochore a constitutive Cdc5 localization can produce a growth defect and dissect which subunits are specifically sensitive to constitutive Cdc5 kinase activity. In addition to the Cdc5-GBP, Cdc5 alone, and cdc5-T242A-PBD*-GBP plasmids, we created an additional kinase-dead mutant (K110A; [50]) and also Cdc5 with mutant PBD (Cdc5-PBD*-GBP). We combined each of these plasmids separately with a subset of the GFP strain collection containing 88 kinetochore and related GFP-tagged proteins and scored the resulting strains for growth as before, but at a higher density of 16 colonies per strain (S2C Fig). Comparing SPI data of Cdc5 (with kinase-dead controls, both T242A and K110A mutants) with kinetochore proteins identifies growth defects with subunits of many key kinetochore subcomplexes including Cse4 (Fig 2E), consistent with a previous finding [27]. Unsurprisingly, the Cdc5-PBD*-GBP had similar effects as Cdc5-GBP since the GBP-GFP association bypasses the function of PBD binding (S2C Fig). As Polo kinase can activate the SAC in other organisms [51–58], we assessed whether this was the case in budding yeast. The growth defect caused by association of a SAC kinase (Mps1) to the kinetochore KMN network is rescued by deletion of the gene encoding the SAC component Mad3 [22]. To test whether this is also true for Cdc5, we associated it with multiple kinetochore proteins in *mad3Δ* GFP strains and found the growth defects were not suppressed compared to wild-type cells (S3 Fig). Although we cannot rule out the possibility that the SAC is activated by forced Cdc5 kinetochore recruitment, the growth arrest phenotype is independent of Mad3 and thus the SAC. These data indicate that the kinase activity of Cdc5 is critical for the phenotype of constitutive Cdc5 localization to specific kinetochore subcomplexes.

## Cdc5-Mtw1 association disrupts cell-cycle progression prior to mitosis

Since our Cdc5 kinetochore-specific SPI screen revealed that all the members of the MIND complex (among others) were sensitive to constitutive Cdc5 localization (Fig 2E), and our prior work showed that an Mtw1-Cdc5 association resulted in a strong chromosome instability phenotype [18] we next sought to investigate the Cdc5 association with the MIND complex in

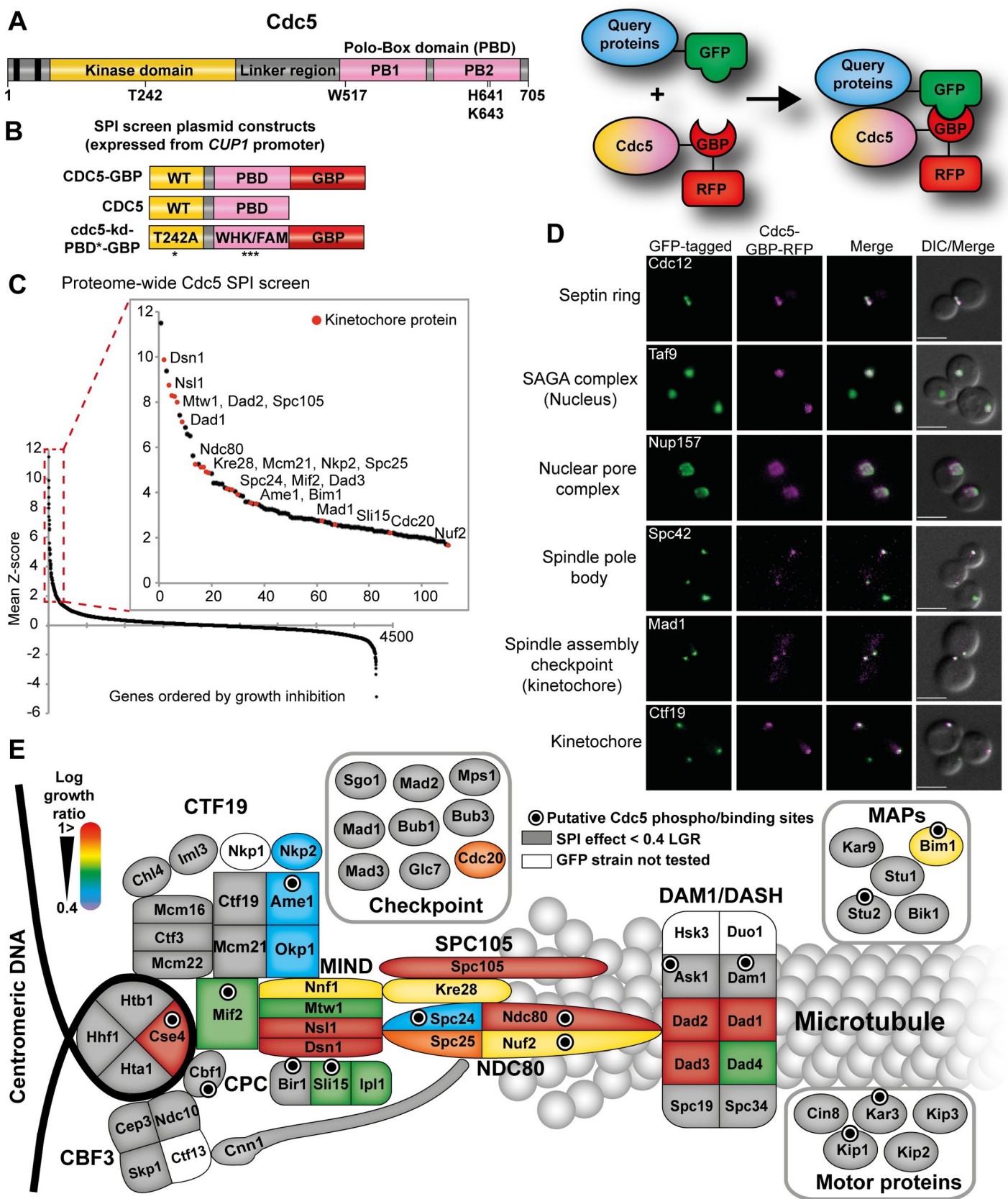

**Fig 2. Proteome-wide Cdc5 SPI screen is enriched for kinetochore proteins.** (A) The key domains of Polo kinase Cdc5 are shown. The N-terminal kinase domain contains a threonine 242 residue when substituted to alanine creates a catalytic-inactive mutant. The C-terminal non-catalytic domain contains two polo-boxes (PB1 and PB2), together called the polo-box domain (PBD) which binds to previously phosphorylated sites which targets the kinase domain of Cdc5 to its substrates and facilitates the phosphorylation at another site on the same substrate or surrounding substrates. A flexible linker region joins the kinase domain and PBD. (B) Schematic of the Cdc5-GBP constructs and controls used in the Cdc5 SPI assays (left). Expression of all constructs are controlled by *CUP1* promoter. The cdc5-kd-PBD*-GBP control contains a mutated form of PBD (W517F, H641A, K643M). The GBP includes an RFP tag. The inset on the right shows a cartoon displaying the Cdc5-GBP interaction with a query GFP-tagged protein. (C) Data from the proteome-wide Cdc5 SPI screen are plotted as in Fig 1A. The inset shows that many kinetochore proteins (red dots) are among the top 100 Cdc5 SPIs. (D) Live imaging of cells containing different GFP-tagged proteins and expressing Cdc5-GBP shows that Cdc5-GBP can be constitutively recruited to many different subcellular locations as judged by colocalization. Scale bars are 5μm. (E) An illustration of the Cdc5 kinetochore SPIs, mapped onto a cartoon representation of the kinetochore. The color-coded map is based on log growth ratios of Cdc5-GBP compared with the average of both kinase-dead controls. The strength of the growth inhibition caused by Cdc5 association is indicated by a color-coded scale with high log growth ratios (LGR > 1) shown in red. Forced interactions that do not produce a growth phenotype are shown in grey (LGR < 0.4). Proteins that contain phosphosites that fit Cdc5 consensus for either phosphorylation or binding (E/D/Q)-X-(pS/T)-X and S-pS/T-P, respectively) are indicated with black dots.

more detail. Moreover, we previously showed that cell-cycle regulated phosphorylation of Cse4 by Cdc5 was important for kinetochore function [27], and thus also asked whether Cdc5 has separate roles at different kinetochore subcomplexes, as has been eluded to in human cells [30,31]. We used a conditional system to induce the Cdc5 kinetochore localization under the control of a *GAL1* promoter. However, overexpression of *CDC5* is lethal in yeast [50], hence we utilized our earlier finding that overexpression of *CDC5* lacking the PBD and fused to GBP (*CDC5ΔC-GBP*) is not [27], presumably because it is unable to bind its substrates (Fig 3A). We showed that overexpression of *CDC5ΔC-GBP* becomes lethal when expressed in GFP-tagged kinetochore strains (Mtw1-GFP and Ndc80-GFP) in contrast to a non-GFP wild-type strain (Fig 3A). Therefore, this allows us to control the Cdc5 associations with different kinetochore complexes in a conditional manner. Moreover, this system can be used to dissect the kinetochore specific function of Cdc5 since *CDC5ΔC* expression does not cause the same pleiotropic effects as the overexpression of wild-type *CDC5*.

Using fluorescence microscopy, we first confirmed that the *GAL1* promoter-controlled expression of Cdc5ΔC-GBP and cdc5ΔC-kd-GBP localized to YFP-tagged kinetochore protein, Mtw1-YFP (Fig 3B). A quantification of the cell-cycle stages after inducing Cdc5ΔC-GBP in an Mtw1-YFP strain also containing a fluorescently tagged tubulin (mTurquoise2-Tub1) revealed that cells accumulated in S-G2-phase as measured by small-budded cells with a single Mtw1 focus or two closely aligned Mtw1 foci and a small tubulin focus (rather than a line; Fig 3C and 3D). This observation contrasts with a forced Cdc5 localization to both Cse4, which caused a strong growth inhibition but did not accumulate cells at any specific stage of the cell cycle [27], and to the DAM1/DASH1 subunit Dad4, which accumulated cells later in the cell

**Table 1. Gene Ontology enrichment of Cdc5 SPIs.**

| GO category Process | Gene name | P-value |
|---|---|---|
| Chromosome segregation | *DSN1, MTW1, NSL1, NDC80, NUF2, SPC24, SPC25, MCM21, NKP2, MIF2, DAD1, DAD2, DAD3, SLI15, SMC2* | 2.4E-10 |
| **Function** | | |
| Microtubule binding | *BIM1, SPC97, SPC98, DAD1, DAD2, DAD3, SPC105* | 8.0E-06 |
| **Component** | | |
| Kinetochore | *DSN1, MTW1, NSL1, NDC80, NUF2, SPC24, SPC25, KRE28, SPC105, MCM21, NKP2, AME1 MIF2, DAD1, DAD2, DAD3, SLI15, MAD1* | 3.4E-15 |
| RNA polymerase II transcription factor complex | *TFB2, MED4, SRB7, ROX3, RGR1, SIN4, TFG1, SPT15, TAF6, TAF12* | 2.7E-06 |
| SAGA complex | *SGF73, SPT20, SGF11, TAF6, TAF12* | 8.8E-05 |

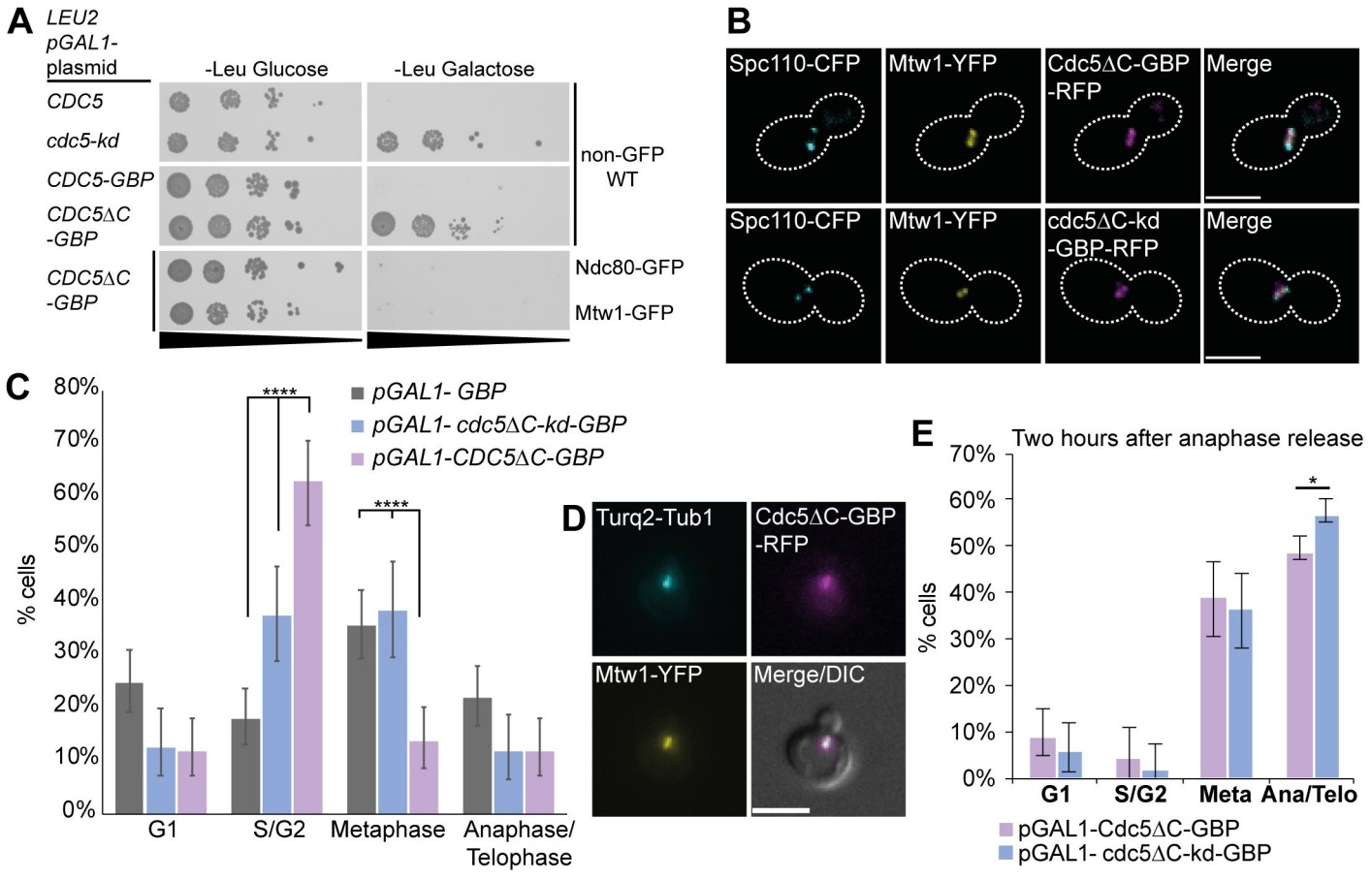

**Fig 3. Forcing Cdc5 association with Mtw1 disrupts cell-cycle progression prior to mitosis.** (A) 10-fold serial-dilution spot assay shows that pGAL1-driven expression of *CDC5* in wild-type (WT) cells is lethal on galactose containing media whereas *CDC5ΔC-GBP* is not. However, expressing *CDC5ΔC-GBP* in GFP-tagged kinetochore strains (Mtw1 and Ndc80) is lethal. (B) Live imaging of cells containing Mtw1-YFP and Spc110-CFP (PT257) and expressing Cdc5ΔC-GBP-RFP (or cdc5ΔC-kd-GBP-RFP) shows that Cdc5-GBP sufficiently colocalizes with YFP foci and also between two SPB foci confirming kinetochore localization. Note that GBP does not have an affinity for CFP. Scale bar represents 5μm. (C) Cell-cycle analysis of Mtw1-YFP cells (T607) containing mTurquoise2-tagged tubulin (Turq2-Tub1) was performed using fluorescence microscopy. Cells from asynchronous cultures containing *pGAL1-CDC5ΔC-GBP* (n = 153), *pGAL1-cdc5ΔC-kd-GBP* (n = 120) or *pGAL1-GBP* (n = 219) were imaged after 4 hours of growth in 2% galactose media to induce expression. Cells that did not show RFP/YFP colocalization were excluded from this analysis. Cells expressing Cdc5ΔC-GBP-RFP were significantly increased in S/G2 phase compared to controls. Fishers exact statistical test; p-values $^{****}$ = p < $10^{-4}$. Error bars indicate 95% binomial confidence intervals (C.I.). (D) Representative cropped image from the analysis in (C) showing a *CDC5ΔC-GBP* expressing cell in S/G2-phase. Scale bar represents 5μm. (E) Mtw1-YFP cells expressing *CDC5ΔC-GBP* (n = 319) during mitosis are able to progress into anaphase/telophase although at a slightly lower frequency than cells expressing *cdc5ΔC-kd-GBP* (n = 211). The metaphase-arrested cells in S5B Fig were released into anaphase by resuspending the cell cultures in galactose media lacking methionine. Two hours after release the cell-cycle stage of these cells was analyzed and revealed that about half of the cells had progressed into anaphase/telophase. Statistical analysis was done using Fishers exact test; p-values $^{*}$ = p < 0.05. Error bars indicate 95% binomial C.I.

cycle, possibly in late anaphase and/or telophase (S4 Fig). Notably, in fission yeast, it has been reported that Polo kinase, Plo1, phosphorylates Dam1 during prometaphase and metaphase for faithful chromosome biorientation [59].

Since the forced Cdc5-Mtw1 interaction resulted in accumulation of cells prior to mitosis, likely in S/G2 phase, we wanted to assess whether the interaction would affect kinetochores in mitosis or prevent cells from progressing from metaphase to anaphase. For this, we used a methionine-dependent Cdc20-depletion system [60,61] to arrest cells in metaphase. Before inducing the Cdc5-Mtw1 association we depleted Cdc20 for two hours which arrested ~70% of the cells in metaphase (S5A and S5B Fig). We then induced either Cdc5ΔC-GBP or cdc5ΔC-kd-GBP in Mtw1-YFP cells and captured images after further two hours. We noticed that the length of the mitotic spindle was reduced by induction of Cdc5ΔC-GBP as quantified by the

inter-sister-kinetochore distance (mean value with Cdc5ΔC-GBP of 0.62μm compared to 0.89μm with cdc5ΔC-kd-GBP; S5C Fig).

Cdc5 localizes to the centromere/kinetochore during mitosis [26,27] thus is unlikely to disrupt kinetochore assembly or interactions during this period. However, the reduced mitotic spindle length resulting from the Cdc5-Mtw1 association in the metaphase-arrested cells may impair anaphase progression. To test this notion, we released the cells from metaphase arrest by Cdc20 restoration and quantified the cell-cycle progression. Two hours after release, Mtw1-YFP cells expressing Cdc5ΔC-GBP continued the cell cycle through anaphase and into telophase, although at a slightly lower frequency than control cells (Fig 3E). These data imply that the short-spindle phenotype caused by the Cdc5-Mtw1 association during metaphase does not prevent anaphase progression, and that sister kinetochores remain attached to the mitotic spindle sufficiently to allow their segregation. Finally, we tested whether the growth phenotype caused by *pGAL1-CDC5ΔC-GBP* expression in Mtw1-YFP cells was dependent on the SAC. However, we found that deleting SAC genes (*MAD1*, *MAD3*, *BUB1* and *BUB3*) was not sufficient to suppress the growth defect (S5D Fig), in agreement with the previous observation (S3 Fig). Collectively, these data suggest that the growth defect caused by the Cdc5-Mtw1 association is not a consequence of a constitutive SAC activation or kinetochore disassembly, rather it can be attributed to premature recruitment of Cdc5 causing pre-metaphase disruptions, most likely in S-, G2-phase or possibly early mitosis, as indicated by the microscopy cell-cycle analysis (Fig 3C and 3D).

## The Cdc5-Ame1 association impairs the outer kinetochore and anaphase progression

It has been shown that Plk1 in human cells interacts with and regulates CENP-U (also known as Polo-Box Interacting Protein 1; PBIP1), the human homolog of Ame1 [62–64]. In yeast, the MIND complex requires the COMA (Ctf19, Okp1, Mcm21 and Ame1) subcomplex of the CCAN for its kinetochore localization and the interaction between these two complexes is mediated by Ame1 [65]. Furthermore, the N terminus of Ame1 contains a number of phosphorylation sites, functions of which remain elusive. The Cdc5 kinetochore-specific SPI screening revealed that COMA complex heterodimer Ame1 and Okp1 (human CENP-UQ), were also sensitive to constitutive Cdc5 localization (Fig 2E).

Since the proteins in the GFP collection are all tagged at the C terminus, we used a different protein association strategy to precisely localize Cdc5 adjacent to the N terminus of Ame1 by generating a direct genetic N-terminal fusion of *AME1* with *CDC5*. As before we decided to omit the PBD to specifically assess the effect of the Cdc5 kinase-domain (*pGAL1-CDC5Δ-C-AME1*), an approach that has been used successfully by others, in both budding yeast and human cells [28–31]. The expression of *CDC5ΔC-AME1* resulted in a strong growth inhibition whereas fusion with the kinase-dead Cdc5 (*pGAL1-cdc5ΔC-kd-AME1*) was indistinguishable from an empty plasmid control (Fig 4A), suggesting that the lethality is not a consequence of N-terminal disruption of Ame1. We tested the cell-cycle progression of cells expressing the *CDC5ΔC-AME1* fusion and found that they accumulated in metaphase compared with cells carrying the empty plasmid control (Fig 4B). However, cells expressing kinase-dead *cdc5ΔC-kd-AME1* accumulated in metaphase to the same extent (Fig 4B). This was unexpected as the kinase-dead fusion did not result in a growth defect. We studied these metaphase cells in more detail and found that cells expressing the *cdc5ΔC-kd-AME1* control had a relatively normal bioriented mitotic spindle (Fig 4C and 4D, white arrows in top panel). In contrast, *CDC5ΔC-AME1* expressing cells often exhibited multiple kinetochore foci (Dad4-YFP) and abnormal spindle as judged by multiple microtubules (Fig 4D). The extra Dad4 foci can be

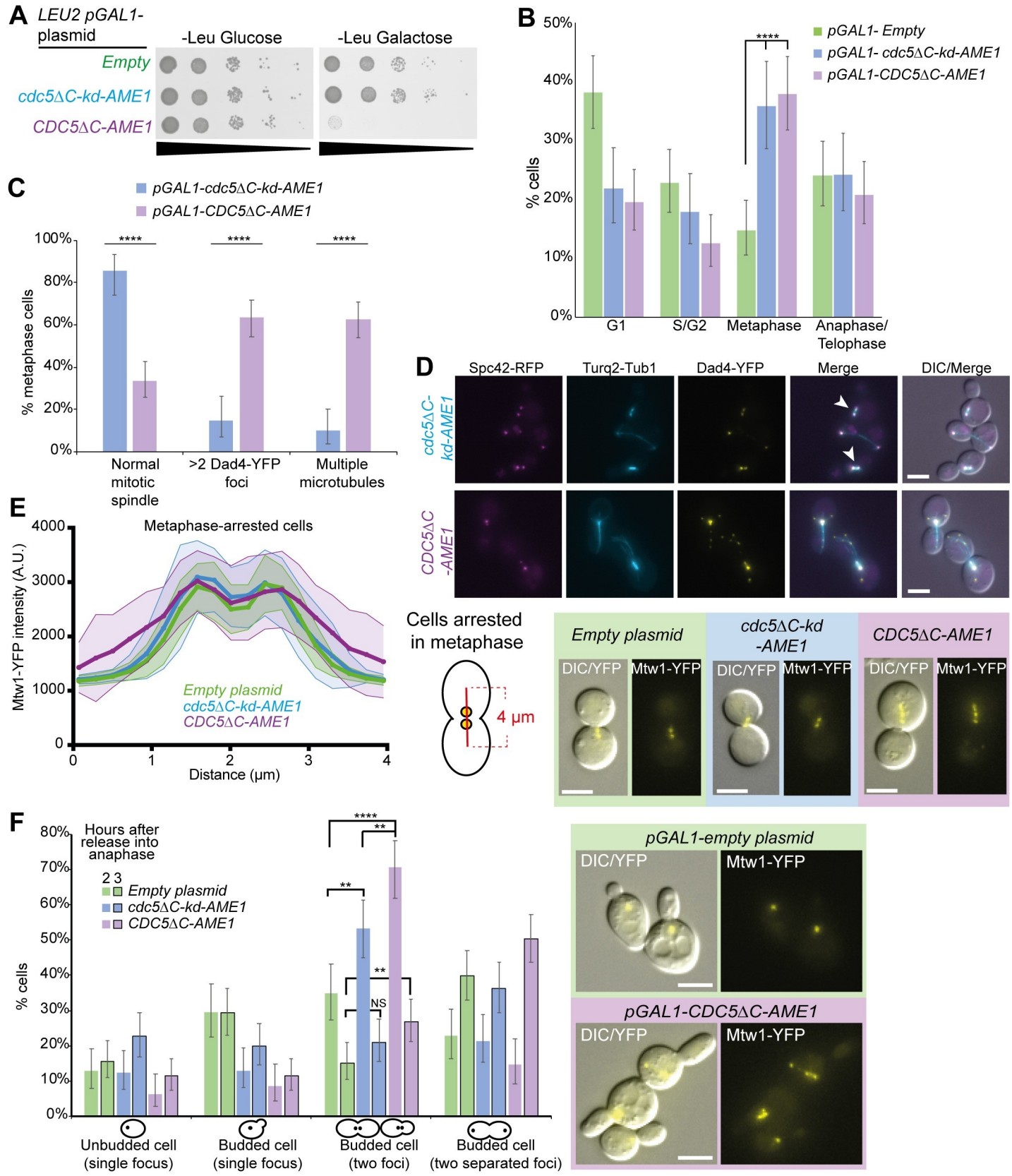

**Fig 4. Association of Cdc5 with Ame1 disrupts Mtw1 foci and mitotic progression.** (A) 10-fold serial-dilution spot assay showing that pGAL1-driven expression of *CDC5ΔC-AME1* prevents growth whereas *cdc5ΔC-kd-AME1* does not. (B) Cell-cycle analysis as in Fig 3C but with Dad4-YFP Turq2-Tub1 cells (T692) also containing RFP-tagged SPB (Spc42-RFP) and expressing *pGAL1-CDC5ΔC-AME1* (n = 223), *pGAL1-cdc5ΔC-kd-AME1* (n = 173) or empty plasmid (n = 249). Cell expressing either *CDC5ΔC-AME1* or *cdc5ΔC-kd-AME1* were significantly increased in metaphase compared to empty plasmid control. Fishers exact statistical test; p-values **** = $p < 10^{-6}$. Error bars indicate 95% binomial confidence intervals. (C) The metaphase cells in (B) were categorized based upon the cell and spindle morphology observed using fluorescence imaging for both *pGAL1-CDC5ΔC-AME1* (n = 93), *pGAL1-cdc5ΔC-kd-AME1* (n = 62). Normal spindle was categorized as two separated Spc42-RFP and Dad4-YFP foci with microtubule signal in between. Cells expressing *CDC5ΔC-AME1* had significantly fewer normal spindles and increased Dad4-YFP foci and microtubules. Fishers exact statistical test; p-values **** = $p < 10^{-4}$. Error bars indicate 95% binomial confidence intervals. (D) Representative cropped images from the analysis in (C) showing *cdc5ΔC-kd-AME1* expressing cells (top) and *CDC5ΔC-AME1* expressing cells (bottom). Normal mitotic spindles are indicated with white arrows. Scale bar indicates 5μm. (E) Mtw1-YFP cells (PT63-12B) were arrested in metaphase using Cdc20 depletion and the Mtw1-YFP fluorescence intensity was quantified along the axis of the mitotic spindle after inducing *CDC5ΔC-AME1* or controls for two hours (see S5A Fig for further description of experimental setup). A 4 μm line with 19 points (illustrated in the inset on the right) was used to include background signal and to cover the spread YFP signal phenotype in cells expressing *CDC5ΔC-AME1* (40 randomly selected mitotic spindles were measured for each condition). The shadowed area indicates standard deviation. Cells expressing *CDC5ΔC-AME1* have a more dispersed arrangement of Mtw1-YFP, as indicated by the flatter profile. Representative cropped images are shown on the right. (F) The metaphase-arrested cells in (E) were released into anaphase and imaged two and three hours later. Two hours after anaphase release (non-outlined bars) cells expressing *CDC5ΔC-AME1* (n = 129) were still arrested in metaphase compared to *cdc5ΔC-kd-AME1* (n = 154) and empty plasmid (n = 149) controls and displayed abnormal Mtw1-YFP foci shown on the right. Three hours after anaphase release (black-outlined bars) there was no statistical difference between cells expressing *CDC5ΔC-AME1* (n = 220) and *cdc5ΔC-kd-AME1* (n = 190) but cells containing empty plasmid control (n = 198) showed significantly reduced number of budded cells with two kinetochore foci compared to *CDC5ΔC-AME1*. Fishers exact statistical test; p-values ** = $p < 0.005$, **** = $p < 10^{-5}$. Error bars indicate 95% binomial C.I. All scale bars indicate 5μm.

seen at microtubule tips, suggesting they represent kinetochores that remain bound to microtubules but are separated from the kinetochore cluster (Fig 4D, lower panel). This analysis shows that the forced Cdc5-Ame1 association produces a phenotype that differs from the other associations of Cdc5 with Cse4 [27], the MIND complex (Fig 3C and 3D) and the DAM1/DASH complex (S4 Fig). As with the Cdc5-Mtw1 association we also found that deleting a SAC component (*mad3Δ*) did not relieve the *CDC5ΔC-AME1* growth phenotype (S6A Fig). Finally, we also investigated *CDC5ΔC-AME1* expressing cells by testing the DNA content using flow cytometry. Compared to both an empty plasmid control and *cdc5ΔC-kd-AME1* expression, cells expressing *CDC5ΔC-AME1* had less cells with a complete 2C DNA content (S6B Fig). Thus, despite their metaphase spindle appearance, it appears that *CDC5ΔC-AME1*-expressing cells are failing to complete full DNA replication or have an abnormal DNA content. Together these data show that cells expressing *CDC5ΔC-AME1* have aberrant kinetochore foci consistent with impaired metaphase to anaphase transition and also lower DNA content.

Since the forced Cdc5-Mtw1 association during mitosis mildly affected the transition from metaphase to anaphase/telophase despite exhibiting a short spindle phenotype (Fig 3E and S5C Fig), we next asked whether this was the case for the Cdc5-Ame1 association. Also, since both *CDC5ΔC-AME1* and *cdc5ΔC-kd-AME1* expression lead to accumulation in metaphase we wanted to assess their ability to transition through mitosis. We induced *CDC5ΔC-AME1* and *cdc5ΔC-kd-AME1* expression in metaphase-arrested cells and released them into anaphase using the Cdc20-depletion system as before. Prior to anaphase release, both the control cells containing an empty plasmid and the *cdc5ΔC-kd-AME1*-expressing cells were arrested in metaphase with the Mtw1-YFP forming two separated foci as expected for bioriented spindles, whereas cells expressing *CDC5ΔC-AME1* often had multiple Mtw1-YFP foci or a scattered signal spread along the mitotic spindle (Fig 4E). Since the kinetochore signal frequently exhibited multiple foci, we deemed it inappropriate to quantify sister-kinetochore distance in these cells as we did for the forced Cdc5-Mtw1 interaction in S5C Fig. This declustered or scattered kinetochore phenotype caused by *CDC5ΔC-AME1* induction was striking and differed from the cells expressing the *cdc5ΔC-kd-AME1* fusion, suggesting the phenotype is specific to the kinase activity of Cdc5 and not Ame1 N-terminal fusions. Furthermore, two hours after release from metaphase arrest about 70% of *CDC5ΔC-AME1* expressing cells remained in metaphase compared to 50% of *cdc5ΔC-kd-AME1*-expressing cells and ~35% of control cells containing an

empty plasmid (Fig 4F). The *CDC5ΔC-AME1*-expressing cells also exhibited compromised kinetochores as judged by multiple Mtw1-YFP foci potentially explaining their delayed progression through mitosis (Fig 4F, right panel). However, three hours after metaphase release most of the *CDC5ΔC-AME1* expressing cells had progressed into anaphase, but with ~30% still remaining in metaphase, whereas both control cells and the *cdc5ΔC-kd-AME1*-expressing cells had continued through mitosis to a greater extent (Fig 4F), consistent with expression of *cdc5ΔC-kd-AME1* not interfering with anaphase progression. The multiple Mtw1-YFP foci in *CDC5ΔC-AME1* cells, made it challenging to distinguish metaphase from anaphase cells (S6C Fig, bottom panel). Nevertheless, the analysis indicates that about half of the cells expressing *CDC5ΔC-AME1* did not retain metaphase arrest and eventually initiated metaphase to anaphase transition, but with aberrant Mtw1-YFP foci (Fig 4F and S6C Fig). To test whether the effect of *CDC5ΔC-AME1* was specifically affecting the MIND and DAM1/DASH complexes at the outer kinetochore, we also assessed the *CDC5ΔC-AME1* phenotype in other GFP-tagged kinetochore strains. Inducing *CDC5ΔC-AME1* in asynchronized cell cultures resulted in ~30% of Ndc80-GFP cells exhibiting a scattered or fragmented Ndc80-GFP phenotype, similar to Mtw1-YFP, compared to ~7% of *cdc5ΔC-kd-AME1* control cells (S7A and S7B Fig). We also investigated other components of the CCAN and found the scattered kinetochore phenotype was minimal for Okp1-GFP (S7C and S7D Fig) and absent for Ctf19-GFP (S7E and S7F Fig) and Mif2-GFP (S7G and S7H Fig). This indicates that outer kinetochore proteins, such as subunits of the DAM1/DASH complex and KMN network are mislocalized by expression of *CDC5ΔC-AME1* while inner kinetochore CCAN components are less affected.

Ame1 has not been shown to be a substrate for Polo kinase in budding yeast, in contrast to CENP-U in human cells [62–64]. However, Ame1 does contain multiple N-terminal phosphoserines [66–69], many of which are consistent with CDK and Cdc5 consensus sites that have not yet been characterized (S8A Fig). To test whether the growth defect caused by *CDC5ΔC-AME1* was due to constitutive phosphorylation of these sites, we mutated them to alanines (*pGAL1-CDC5ΔC-ame1-7A*) and asked whether inhibiting phosphorylation of Ame1 would suppress the growth defect. However, these mutations were not sufficient to suppress the growth phenotype (S8B Fig). An alternative possibility is that the growth defect is a consequence of Cdc5 phosphorylating additional sites on neighboring proteins. For instance, the N-terminal tail of Cse4 contains Cdc5 phosphosites [27] and interacts with the COMA complex [2,70,71] thus providing a possible route for *CDC5ΔC-AME1* to constitutively phosphorylate Cse4, which we found previously to mislocalize Cse4 and inhibit growth [27]. We tested this notion by expressing the *CDC5ΔC-AME1* fusion in a strain containing a phospho-deficient mutant of Cse4 (*cse4-9A*; [27]). However, the prevention of Cse4 phosphorylation did not suppress the growth defect (S8C Fig), suggesting the phenotype is not a consequence of indirectly targeting Cdc5 to the Cse4 phosphosites via the N terminus of Ame1. In summary, these findings show that the constitutive Cdc5-Ame1 interaction impairs kinetochore function and mitotic progression. This contrasts with the Cdc5-Mtw1 interaction which showed a pre-mitotic defect and did not prevent metaphase to anaphase progression. Despite exhibiting impaired kinetochores, the metaphase-arrested cells expressing *CDC5ΔC-AME1* do not remain arrested in metaphase suggesting a failure to activate or maintain a SAC response. Additionally, we show this phenotype is not individually dependent upon hyperphosphorylation of Ame1 or Cse4.

## Identifying suppressors of the forced Cdc5-Ame1 association phenotype

To further understand the function of Cdc5 as a possible regulatory factor at the interface of the COMA and MIND complexes, we performed a genome-wide suppressor screen to study

the genetic dependencies on the *CDC5ΔC-AME1* growth phenotype. We introduced the *CDC5ΔC-AME1* fusion plasmid and, separately, *CDC5* alone and an empty vector as controls, all under the control of the *GAL1* promoter, into the gene deletion collection of non-essential genes (~4800 deletion strains; [72]) and the temperature-sensitive mutant (ts) collection of essential genes (787 strains with ts mutations of 497 genes; [73]) (Fig 5A). Gene ontology (GO) enrichment analysis of the total *CDC5ΔC-AME1* suppressors (287 geneΔ mutants and 43 ts mutants; 330 genes in total; S9A–S9D Fig and S4 Data) showed enrichment for genes involved in intracellular transport, histone deacetylation, mediator complex and RNA metabolism (specifically non-coding RNA metabolic process, RNA methyltransferase activity and RNAPII transcription elongation; Table 2). The *CDC5* overexpression suppressors were enriched for intracellular transport, 7-methylguanosine RNA capping and dynein complex (Table 2). Importantly, ts mutation of Cdc5 (*cdc5-1*; [74]) suppressed the *CDC5* overexpression but not the *CDC5ΔC-AME1* fusion phenotype (S9E Fig). The only GO enrichment category identified that was specific for *CDC5ΔC-AME1* suppressors (94 geneΔ mutants and 33 ts mutants; 127 genes in total (S9A–S9D Fig) were genes involved in regulation of histone modification (*SPN1*, *XBP1*, *SGV1*, *VPS71*, *SPT6*; p-value = $1.85 \times 10^{-4}$). Collectively, these data show that deletions or mutations of genes involved in transcription and/or chromatin remodeling can suppress the growth defect caused by the constitutive Cdc5-Ame1 interaction.

## The Cdc5-COMA association phenotype requires a functional Spt4/5 complex

One of the strongest suppressors of *CDC5ΔC-AME1* was a deletion of *SPT4* (and *YGR064W*, an uncharacterized ORF that partially overlaps with *SPT4*) (Fig 5B). Spt4 is a universally conserved part of the DSIF complex (also known as Spt4/5 complex), along with Spt5, which is involved in transcription elongation in all domains of life [75]. The *CDC5ΔC-AME1* was also suppressed by *spt5-194* ts mutation at 26.5°C (Fig 5B). Interestingly, Spt4 is required for faithful chromosome segregation [76] and localizes to kinetochores [77]. Furthermore, the Spt4/5 complex couples the transcription elongation activities with pre-mRNA processing and chromatin remodeling [78,79]. We also found other *CDC5ΔC-AME1* suppressor mutations in genes that function in the same pathway as the Spt4/5 complex, for example the histone chaperone Spt6 (*spt6-14*; Fig 5B) and its interaction partner the chromatin remodeling factor Spn1 (also known as Iws1; *spn1-K192N*; S9I Fig). Furthermore, we identified a mutant of the Sgv1 kinase (also known as Bur1; homolog of human CDK9), as a *CDC5ΔC-AME1*-specific suppressor (both *sgv1-35* and *sgv1-80*; Fig 5B). Sgv1 phosphorylates both Spt5 and RNAPII (Rpo21) to facilitate transcription elongation and recruitment of Spt6 [78,80–82]. Another complex with a role in transcription elongation and kinetochore function is the SWI/SNF chromatin remodeling complex, RSC [83–87]. We found that deletions of the RSC complex subunits, *NPL6*, *LDB7* and to a lesser extent *RSC1*, also suppressed the *CDC5ΔC-AME1* fusion (S9G Fig and S4 Data). These data show that deletions or mutations of specific genes that interact in the same pathway and have roles in histone modification, transcription elongation and mRNA processing as well as in kinetochore function, suppress the growth phenotype caused by expression of *CDC5ΔC-AME1*.

Since it has been reported that Spt4 associates with centromeres in a RNAPII-independent manner and optimal Cse4 kinetochore localization requires Spt4 [77], we sought to investigate the *spt4Δ* suppressor in more detail. First, we confirmed that the growth defect caused by *CDC5ΔC-AME1* fusion, as well as *CDC5* overexpression, was completely rescued by *spt4Δ* (Fig 5C). Next, since our SPI assays found that a constitutive Cdc5 kinase activity at many different kinetochore subcomplexes produced growth phenotypes (Fig 2E), we asked whether the *spt4Δ*

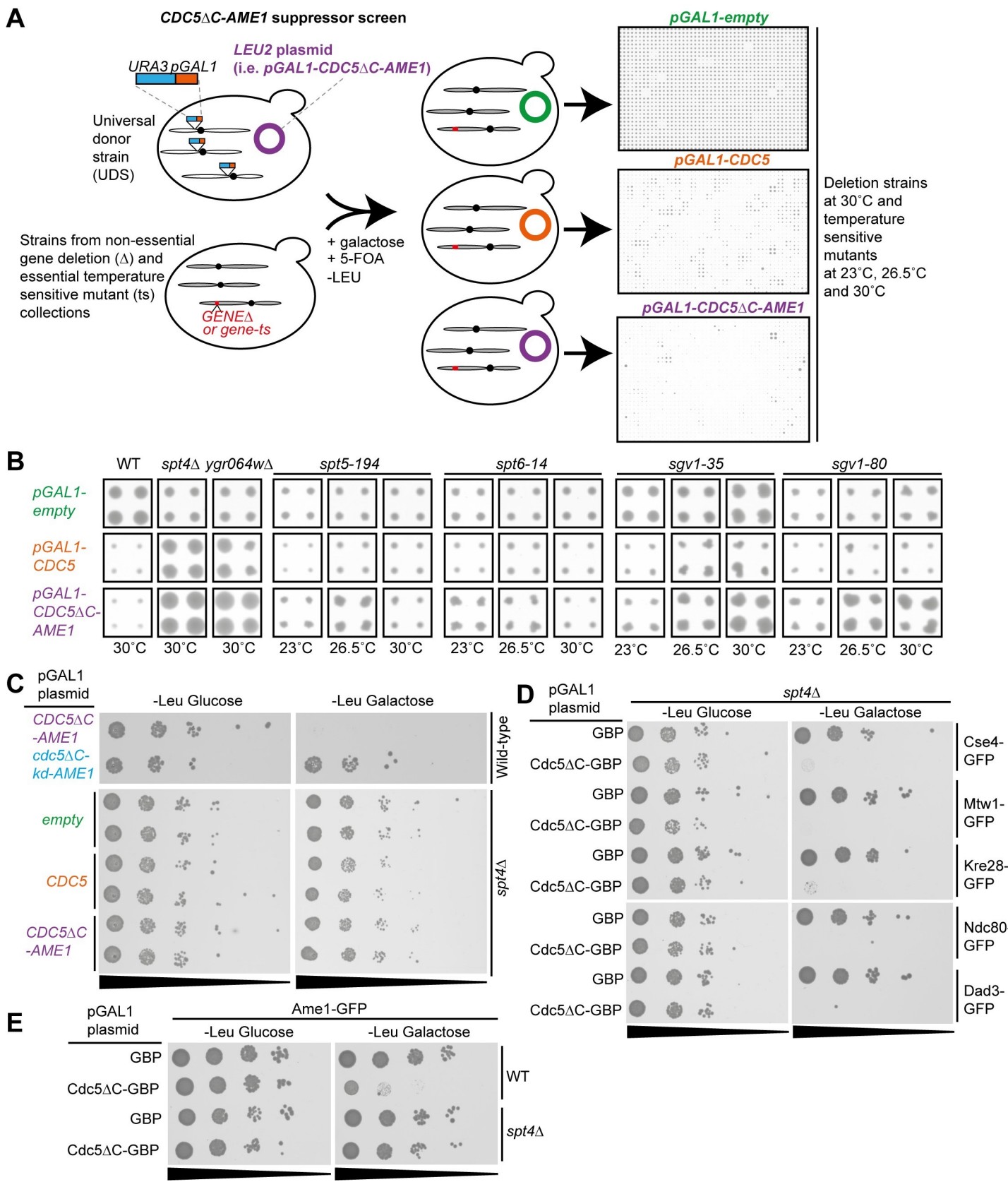

**Fig 5. Suppressors of the forced Cdc5-Ame1 interaction phenotype.** (A) Diagram showing the experimental layout of the suppressor screen. Selective ploidy ablation (SPA) was used to introduce *pGAL1-CDC5ΔC-AME1*, *pGAL1-CDC5* and *pGAL1-empty* plasmids into the deletion (Δ) and temperature-sensitive mutant (ts) collections. The Δ screen was performed at 30˚C and the ts screen at 23˚C, 26.5˚C and 30˚C. The agar plates from the final selection step were scanned after three days and representative examples are shown (see methods for further details). (B) A selection of cropped images of colonies from the suppressor screens showing that growth defects caused by *pGAL1-CDC5ΔC-AME1* or *pGAL1-CDC5* are suppressed by genetic deletions or mutations. (C) 10-fold serial dilutions spot assay with wild-type and *spt4Δ* strains containing *pGAL1-CDC5ΔC-AME1* and control plasmids shows that *spt4Δ* suppresses growth defect caused by *CDC5ΔC-AME1* expression. (D) 10-fold serial dilutions spot assay with GFP-tagged kinetochore strains (Cse4-GFP (internally tagged), Mtw1-GFP, Kre28-GFP, Ndc80-GFP and Dad3-GFP) containing *pGAL1-CDC5ΔC-GBP* and *pGAL1-GBP* plasmids shows that spt4Δ is not sufficient to suppress the growth defect caused by constitutive localization of Cdc5 to these proteins. (E) 10-fold serial dilutions spot assay with wild-type and *spt4Δ* strains encoding Ame1-GFP and *pGAL1-CDC5ΔC-GBP* and *pGAL1-GBP* plasmids showing that *spt4Δ* suppressed growth defect caused by constitutive Cdc5 association with Ame1.

suppression was specific to the Cdc5 association with Ame1. For this purpose, we introduced the *pGAL1-CDC5ΔC-GBP* plasmid into *spt4Δ* strains containing different GFP-tagged kinetochore proteins. To our surprise, none of GFP-tagged kinetochore strains tested (Cse4-GFP internally tagged, Mtw1-GFP, Kre28-GFP, Ndc80-GFP or Dad3-GFP) were suppressed by *spt4Δ* (Fig 5D). We were concerned that the *spt4Δ* rescue of *CDC5ΔC-AME1* growth defect was specific to the genetic fusion construct and not to the forced recruitment of Cdc5 to Ame1. To determine whether this was the case, we introduced the Cdc5ΔC-GBP into Ame1-GFP strains with or without *SPT4*. Remarkably, we found that the growth defect was suppressed by *spt4Δ* (Fig 5E), and similarly for other COMA complex subunits, Okp1 and Mcm21 (S10A and S10B Fig). We note that expression of Cdc5ΔC-GBP in Ame1-GFP cells accumulated cells in metaphase to a similar extent as *CDC5ΔC-AME1*-expressing cells (S10C Fig), and that *AME1* fused to either *CDC5ΔC* or *cdc5ΔC-kd* can replace endogenous Ame1-GFP at the kinetochore (S10D Fig), suggesting it is functional, consistent with previous data [88]. These findings indicate that the phenotype caused by constitutive Cdc5 kinase activity specifically at the COMA complex and not with other kinetochore components requires *SPT4*.

It has been shown that a mutant of Spt4, *spt4-138*, has a more severe chromosome missegregation phenotype than *spt4Δ* [76]. We thus wanted to test the ability of *spt4-138* to suppress *CDC5ΔC-AME1* and found that it did suppress to the same extent as *spt4Δ* (S10E Fig). These data indicate that inhibition of Spt4 function is sufficient to suppress the growth defect caused

**Table 2. Gene Ontology enrichment of suppressors of *CDC5ΔC-AME1* and *CDC5*.**

| *CDC5ΔC-AME1* suppressors: | | |
|---|---|---|
| **GO terms** | **Gene name** | **P-value** |
| Intracellular transport | *PAC1, PAC11, SPC72, DYN1, AIM21, ARP1, GPA1* | 3.9E-04 |
| Histone deacetylation | *RXT3, HDA1, SET3, SIF2, HOS2, HOS4, RCO1* | 6.8E-04 |
| RNA metabolism | 53 gene deletions or ts mutants* | 5.2E-04 |
| Mediator complex | *SRB5, MED4, CSE2, NUT1, BUD27, SOH1, MED8* | 6.8E-04 |
| Set3 complex | *HOS2, HOS4, SET3, SIF2* | 9.0E-04 |
| *CDC5* suppressors: | | |
| **GO terms** | **Gene name** | **P-value** |
| intracellular transport | *KIP2, PAC1, PAC11, SPC72, DYN1, AIM21, ARP1, DYN3, NUM1, NDL1* | 2.3E-05 |
| 7-methylguanosine RNA capping | *CTL1, SPT4, CEG1, STO1, TGS1* | 7.5E-04 |
| Dynein complex | *DYN3, PAC11, JNM1, DYN1, ARP1* | 9.0E-04 |

*See S4 Data for list of genes.

by *CDC5ΔC-AME1*. Furthermore, we assessed whether the Sp4/5 complex was affected by the Cdc5-Ame1 association and found that the Spt4-GFP signal was increased upon induction of *CDC5ΔC-AME1* compared to *cdc5ΔC-kd-AME1* (S9F Fig), and similarly for Spt5-GFP, although to a lesser extent (S9G Fig).

It is possible that the Spt4 function is required for transcription driven by the *GAL1* promoter. Thus, the *spt4Δ* mutant would rescue the lethal phenotype of *CDC5ΔC-AME1* by ablating its expression. However, we note *spt4Δ* does not rescue pGAL1-driven expression of *CDC5ΔC-GBP* in GFP-tagged kinetochore strains, except for strains with GFP-tagged COMA subunits (Fig 5D–5E and S10A and S10B Fig), strongly indicating that *spt4Δ* mutants do not act by simple down-regulation of pGAL1 driven expression. We also assessed this hypothesis by asking whether *spt4Δ* was able to suppress *pGAL1*-driven overexpression of other genes known to disrupt growth, *PSH1*, *BIM1 and BIK1* [89,90]. However, the slow growth phenotype of cells overexpressing *PSH1* was not suppressed by *spt4Δ* (S11A Fig, top panel) and cells lacking *SPT4* and overexpressing *BIM1* or *BIK1* were completely inhibited for growth to the same extent as wild-type cells (S11A Fig, bottom panel). Finally, we asked whether the rescue by *spt4Δ*, could be reverse-complemented by restoring expression of the *SPT4* gene. We introduced plasmids containing *SPT4*, either under the control of *GAL1* or *CUP1* promoter, into *spt4Δ* strains expressing *CDC5ΔC-AME1*. In both cases, the rescue of the *CDC5ΔC-AME1* phenotype by *spt4Δ* was repressed when *SPT4* was reintroduced (S11B Fig). Collectively, these data indicate both that the Spt4 protein is required for the *CDC5ΔC-AME1*-dependent growth defect and also that Spt4 is not required to drive expression via the *GAL1* (or *CUP1*) promoter. This latter conclusion is supported by work showing that transcription of most genes is not affected by *spt4Δ*, and as an elongation factor, Spt4 is highly specific for transcription of repetitive DNA located at ORFs or non-coding regions, with A-rich repeats showing the strongest dependence on Spt4 [91]. Finally, we note that deletion of *GAL4*, the transcription factor required for activating galactose genes, was a much weaker suppressor of *CDC5ΔC-AME1* than *spt4Δ*. To summarize, these data show that Spt4 is not required for pGAL1-driven expression and suggest that *spt4Δ* rescues phenotypic effects of *pGAL1-CDC5ΔC-AME1* association rather than preventing its expression.

## The forced Cdc5-Ame1 association increases CEN RNA levels which largely depends on Spt4

Recent studies found that a deletion of the gene encoding the CDEI-binding protein, Cbf1, increased centromeric, non-coding transcripts and chromosome missegregation [92,93]. Although our *CDC5ΔC-AME1* suppressor screen identified mutants of transcription elongation and mRNA processing genes, we felt that Cdc5 was unlikely to be directly affecting centromeric (CEN) transcription for various reasons. First, although deletion of Cbf1 increases CEN transcription, this is not lethal [92,93]. Second, a deletion of Cbf1 is synthetic lethal with a number of kinetochore mutants [94], which suggests that increased CEN transcription could be a consequence of kinetochore disruption. Third, although Polo kinase has been shown to have multiple kinetochore targets in many organisms, it has not been shown to be involved in transcription at the centromere. We suspected the *CDC5ΔC-AME1* fusion could be destabilizing the kinetochore, making it more sensitive to endogenous levels of CEN transcription. Hence, the deletion of transcription elongation factors such as Spt4 may result in desensitized kinetochores that are able to cope with unstable interactions. Thus, we wanted to determine whether the inhibition of transcription elongation was sufficient to suppress the *CDC5ΔC-AME1* growth phenotype in a wild-type genetic background. We tested this by treating the cells separately with two drugs that inhibit transcription elongation, 6-Azauracil

(6-AU) and mycophenolic acid (MPA) [81,95–97]. We did not observe suppression of the *CDC5ΔC-AME1* growth defect by 6-AU or MPA treatments (S12A–S12C Fig). This suggests that the *CDC5ΔC-AME1* phenotype depends on specific proteins involved in transcription elongation or RNA processing, such as a functional Spt4/5 complex, rather than general elongation activity. This also contradicts the idea that the *CDC5ΔC-AME1* fusion destabilizes the kinetochore which can cope when general transcription elongation is inhibited. Notably, we did not identify many transcription factors as strong suppressors of *CDC5ΔC-AME1* nor did we find that a deletion of *DST1*, encoding the general elongation factor TFIIS (which is non-essential in *S. cerevisiae)*, suppressed the growth defect of *CDC5ΔC-AME1*. Together these data indicate that inhibiting transcription elongation is not sufficient to suppress the *CDC5ΔC-AME1* phenotype.

CENP-A deposition in human cells requires Plk1 phosphorylation of the Mis18 complex [98,99], which budding yeast lacks. Moreover, it has been shown that CEN transcription and a specific level of CEN RNAs are important for CENP-A deposition [99–103] and kinetochore function (reviewed in [104,105]). Cse4$^{CENP-A}$ deposition in budding yeast occurs in S-phase after replication of the CEN DNA [6]. Recently, it was shown that Cbf1 is involved in restraining CEN transcription to late S-phase after CEN replication [92,93]. We asked whether Cse4 localization was affected by the Cdc5 association with the COMA complex and examined Cse4 foci in cells expressing *CDC5ΔC-AME1*. However, Cse4 was not mislocalized and the levels were slightly elevated in response to *CDC5ΔC-AME1* (S13A Fig). This contrasts with direct association of Cdc5 to Cse4, which results in Cse4 mislocalization [27]. We also examined Cse4 foci in cells that were arrested in metaphase (using Cdc20 depletion) and found that *CDC5ΔC-AME1* cells frequently had a single, brighter Cse4-YFP focus (S13B and S13C Fig). Collectively, these data imply the *CDC5ΔC-AME1* fusion causes disruption at the centromere which declusters or scatters outer kinetochore complexes along the spindle, whereas inner kinetochore proteins such as Cse4 collapse into a single focus.

The centromere is transcribed into non-coding RNAs in a highly regulated process and both reduced and elevated CEN RNA levels can disrupt kinetochore function and are associated with human disease such as cancer (reviewed in [104,105]). To ask whether the constitutive Cdc5 association with the COMA complex was influencing CEN RNA levels we performed reverse transcription PCR (RT-PCR). Compared to an empty plasmid control, levels of CEN RNA from chromosomes I, III and VIII (CEN1, 3 and 8) were increased by induction of *CDC5ΔC-AME1* for five hours (Fig 6A). In addition, we found that the increased CEN1 and CEN3 RNA was dependent on *SPT4*, whereas CEN8 was not (Fig 6A). It has been proposed that centromeres are differentially transcriptionally regulated [93], which may explain the difference between CEN1/3 and CEN8. These results indicate that association of Cdc5 to the COMA complex can upregulate CEN RNA levels, which largely depends on the elongation and mRNA-processing factor Spt4. To test whether changes in CEN RNA underlie the kinetochore phenotypes seen by the forced Cdc5 recruitment, we next investigated whether the declustered kinetochore phenotype (seen in Fig 4E and 4F) were connected to the increased CEN RNA by repeating the fluorescence microscopy experiments in a *spt4Δ* background. Strikingly, the scattered kinetochore phenotype was completely absent in *spt4Δ* cells (Fig 6B). Likewise, the severe Mtw1-YFP phenotype observed after the release into anaphase (S6C Fig) was dependent on *SPT4*, as *spt4Δ* cells expressing *CDC5ΔC-AME1* were indistinguishable from controls (S14 Fig).

We were surprised to find that expression of *CDC5ΔC-AME1* resulted in accumulation of CEN transcripts and that *spt4Δ* suppressed this to a large extent and also completely relieved the scattered kinetochore phenotype. To confirm whether these phenotypes were connected and whether a deletion of *SPT4* could suppress a phenotype associated with increased CEN

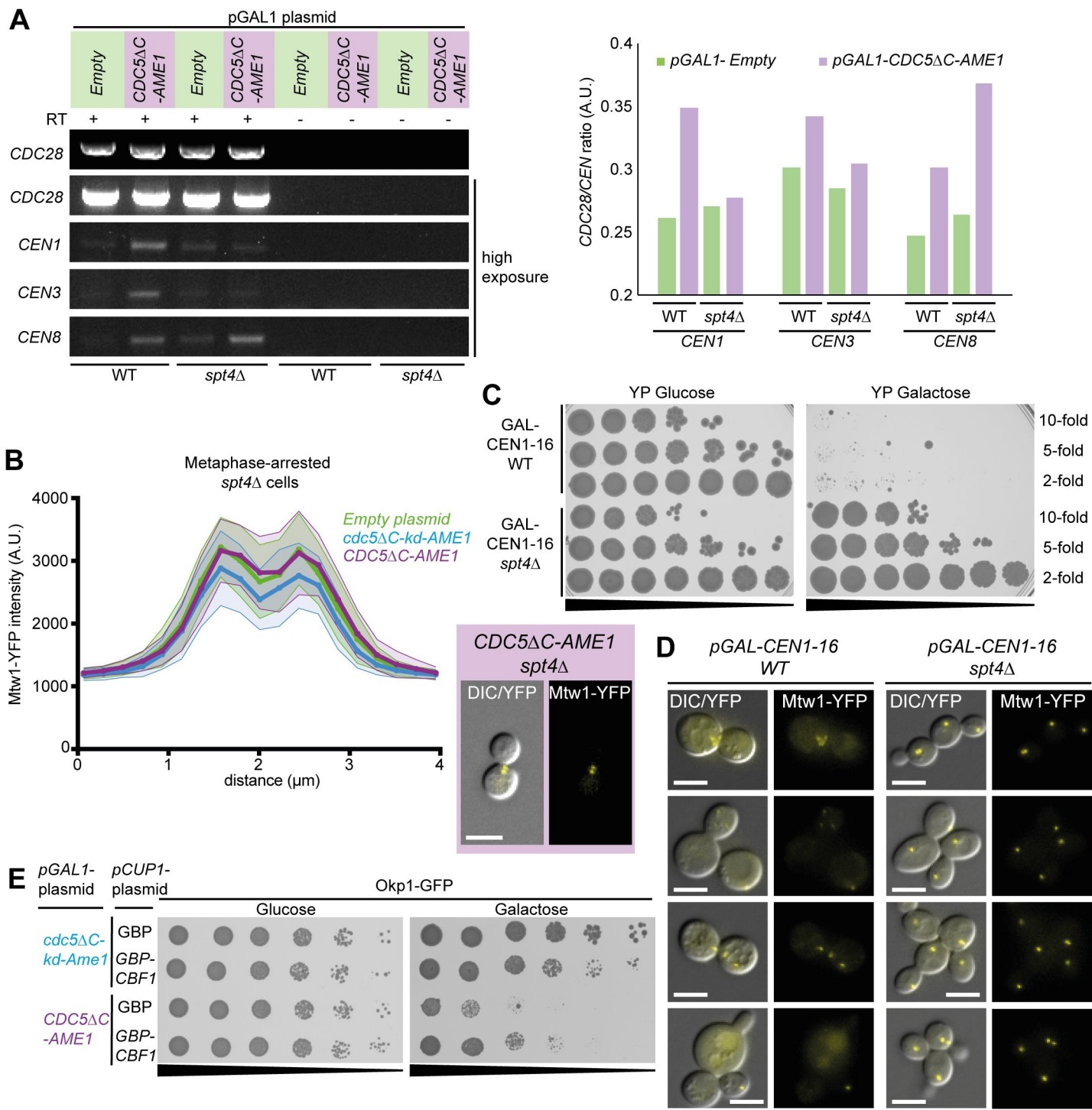

**Fig 6. Association of Cdc5 with Ame1 increases centromeric RNA levels which largely depends on Spt4.** (A) Reverse transcription PCR (RT-PCR) was performed after five hours of induction of *CDC5ΔC-AME1* in wild-type and *spt4Δ* cells. Reactions without reverse transcription were used to control for potential contaminating DNA. Due to very faint CEN signal the exposure was increased to visualize the CEN RNA bands (see Materials and methods for further details). Quantification of the relative CEN RNA levels is shown in the graph on the right. Ratio of the *CDC28* RNA control to the *CEN* RNAs was calculated. (B) Live imaging of metaphase-arrested Mtw1-YFP cells expressing *CDC5ΔC-AME1* or controls and the kinetochore foci analysis was performed as in Fig 4 but in a *spt4Δ* strain (T739). The scattered Mtw1-YFP foci phenotype seen in Fig 4A was absent in *spt4Δ* cells (represented in the inset on the right). (C) A serial dilutions spot assay (10-, 5- and 2-fold) was performed with a GALCEN1-16 strain in both wild-type (W8164-2B) and *spt4Δ* background (T682). GALCEN1-16 *spt4Δ* strain grew equally well on glucose and galactose after two days at 30˚C, in contrast to GALCEN1-16 WT which did not grow on galactose. (D) GALCEN1-16 Mtw1-YFP strains in either a WT (T360) or *spt4Δ* (T738) background were grown in galactose media for 16 hours and then imaged with fluorescence microscopy to investigate the kinetochore phenotype. In WT cells we

found a range of phenotypes with cells showing scattered/fractured kinetochores or some with a diffused Mtw1-YFP signal whereas in *spt4Δ* cells this was not the case. We note that after 4–6 hours in galactose the cells did not show a clear phenotype. All scale bars are 5μm. (E) 5-fold serial dilutions spot assay of Okp1-GFP cells coexpressing *CDC5ΔC-AME1* and *GBP-CBF1* or *GBP* shows that additional Cbf1 recruitment partially suppresses the growth defect caused by *CDC5ΔC-AME1* expression.

RNA independently of Cdc5, we decided to investigate a strain that has been designed to conditionally transcribe centromeres via *GAL1* promoters adjacent to every centromere [106–108]. We asked whether Spt4 contributed to lethality of the GALCEN1-16 strain on galactose medium, caused by persistent transcription and subsequent destabilization of all the centromeres. Strikingly, the lethality of the GALCEN1-16 strain was completely rescued by *spt4Δ* and the strain was able to grow equally well on galactose and glucose media (Fig 6C). In addition, using fluorescence microscopy we found that *spt4Δ* rescued kinetochore disruption associated with growing GALCEN1-16 cells in the presence of galactose (Fig 6D). These findings show that Spt4 is required for the aberrant kinetochore phenotype in both *CDC5ΔC-AME1* expressing cells and GALCEN1-16 strain.

Since Cbf1 was recently identified as a repressor of CEN transcription [92,93], we hypothesized if the *CDC5ΔC-AME1* fusion results in hyperactivation of CEN transcription then overexpression of *CBF1* may suppress the growth phenotype. However, *CBF1* overexpression driven by either the *CUP1* (in the presence of copper to increase expression) or *GAL1* promoters were not sufficient to suppress the *CDC5ΔC-AME1* growth defect (S15A and S15B Fig). Since, it is possible that overexpression of *CBF1* might not be adequate to overcome the constitutive Cdc5 activity resulting from the forced COMA complex localization. We tethered Cbf1 to the COMA complex by expressing GBP-tagged Cbf1 (*pCUP1-GBP-CBF1*) in an *OKP1-GFP* strain while inducing the *CDC5ΔC-AME1* fusion. We found that additional recruitment of Cbf1 did partially mitigate the growth defect of *CDC5ΔC-AME1* (Fig 6E). Since, Cbf1 represses CEN transcription, these data suggest that the *CDC5ΔC-AME1* growth defect is at least in part due to elevated CEN RNA levels.

## Discussion

Here, we have manipulated Cdc5 regulation in both space and time in a controlled manner. Our data show that the kinetochore is especially sensitive to constitutive Cdc5 recruitment. We find that Cdc5 recruitment to separate kinetochore subcomplexes causes different phenotypes. This supports the notion that Polo kinase has multiple roles at the kinetochore [30]. The forced Cdc5-MIND interaction caused a pre-mitotic arrest (Fig 3C and 3D) but did not prevent anaphase progression when induced during mitosis (Fig 3E). In contrast, the forced Cdc5-COMA interaction accumulated cells in metaphase (Fig 4B–4D) and when induced during mitosis caused a scattered outer kinetochore phenotype and impaired anaphase progression (Fig 4E and 4F). Although it is known that Cdc5 localizes at the centromere/kinetochore during mitosis, the precise binding substrates of Cdc5 have not been fully characterized. Snead et al. (2007) found that the PBD of Cdc5 interacts with both Cse4, in agreement with our prior work [27], and Ndc80 supporting the notion that Cdc5 localizes both at the inner and the outer kinetochore. They also showed that anaphase is impaired when Cdc5 activity is diminished, emphasizing that Cdc5 is important for anaphase progression [25]. Our data suggest that forced Cdc5 recruitment to the kinetochore arrests cell growth in a SAC-independent manner. Polo kinase in many metazoans is involved in the SAC by facilitating Mps1 activity. We showed that deletion of the SAC component, *MAD3*, did not suppress the growth defect caused by forced Cdc5 recruitment to many different kinetochore components, including the KMN network, which contrasts with Mps1 SPIs [22]. Although we did also test deletions of

*MAD1*, *BUB1* and *BUB3* in cells with forced Cdc5-Mtw1 association we did not test this in a *MAD2* delete. Since *MAD2* has a genetic interaction with *CDC5* [109–111], it would be worth addressing this in future studies. We note that deletions of all SAC components (including *MAD2*) did not suppress the growth phenotype caused by forced association of Cdc5 with Ame1.

In this work, we focused on the phenotype of constitutive Cdc5 recruitment to the COMA complex, which we show is rescued by mutations of genes involved in transcriptional control and chromatin remodeling. We note that deletions and mutations of genes in other complexes also suppressed the growth defect caused by forced Cdc5-Ame1 association, such as Condensin and Dynein/Dynactin components (S9J Fig and S4 Data), suggesting that other pathways are also involved in the phenotype. Interestingly, Ame1 has a negative genetic interaction with Dynein subunits [36]. In human cells, Plk1 is required for Dynein recruitment to kinetochores [112] and a Dynactin subunit can recruit Plk1 to kinetochores [113]. Our data may indicate that Dynein and Dynactin are also involved in Cdc5 function at the budding yeast kinetochore, which requires further investigation. Here, we focused on the *CDC5ΔC-AME1* suppressor, the transcription elongation and RNA-processing factor Spt4, and provide evidence that the constitutive Cdc5-COMA recruitment results in increased CEN RNA levels, via a *SPT4*-dependent mechanism (Fig 6A). During the cell cycle, CEN transcription is tightly regulated and is confined to a narrow period in S-phase after the replication of the CEN DNA [93]. Cdc5 is expressed in S-phase and may be present at the centromere/kinetochore at low levels in late S-phase, and then increasing in G2-phase and mitosis. In prior work, we did not detect Cdc5 interaction with Cse4 in S-phase, however the cells were treated with hydroxyurea which may block CEN replication and thus prohibit CEN transcription. Additionally, we found that induction of the Cdc5-Ame1 association resulted in a lower DNA content despite cells exhibiting a metaphase appearance (S6B Fig), suggesting that constitutive Cdc5 localization at the COMA complex and/or premature induction of CEN transcription may interfere with DNA replication.

A role for Cdc5 in regulating CEN RNA levels is highlighted by mapping the Cdc5 SPIs and the *CDC5ΔC-AME1* suppressor data onto the global genetic similarity network, which reveals that 'transcription' and 'chromatin' are the two main functional classes that overlap in these two datasets (Fig 7A). Furthermore, relative to the five proteome-wide kinetochore SPI screens, Ctf19, a COMA complex subunit, produced growth defects when associated with proteins involved in transcription (e.g. Ssl2 and Cyc8), mRNA-processing components (e.g. Ceg1) and chromatin (e.g. Set3, Snf2 and Sth1) (Fig 1B and S1 Data).

In human cells, tethering transcriptional activators and silencers to centromeres disrupts kinetochore function [114,115]. If constitutive Cdc5 recruitment to the centromere/kinetochore can increase CEN RNA levels, then we might expect recruitment of the proteins involved in transcription regulation to also give growth phenotypes. Indeed, we found that proteins involved in transcription and histone modification were among the proteins that produced the most consistent growth phenotypes when associated with specific kinetochore proteins (such as Mtw1, Ctf3 and Ctf19; Fig 1C and 1D). For instance, we identified HDAC subunits, such as components of the Set3 and Rpd3 complexes, and members of the transcription factor TFIID and SAGA complexes, Bdf1, Sgf29 and Taf12 (S1C and S1D Fig and S2 Data). In addition, we identified transcriptional regulators (Opi1 and Cyc8), that are involved in recruiting HDACs, the SAGA and the SWI/SNF chromatin remodeling complexes [116–122]. We also identified components of the chromatin-remodeling RSC complex, Rsc58, Rtt102 and Sfh1 as kinetochore SPIs (S2 Data) and that mutants of RSC (*npl6Δ*, *ldb7Δ* and *rsc1Δ*) and HDAC (*set3Δ*, *rxt3Δ*, *hos2Δ* and *hda1Δ*) suppressed the *CDC5ΔC-AME1* phenotype (S9G Fig and S4 Data). These data support the notion that regulated CEN RNA levels and/or

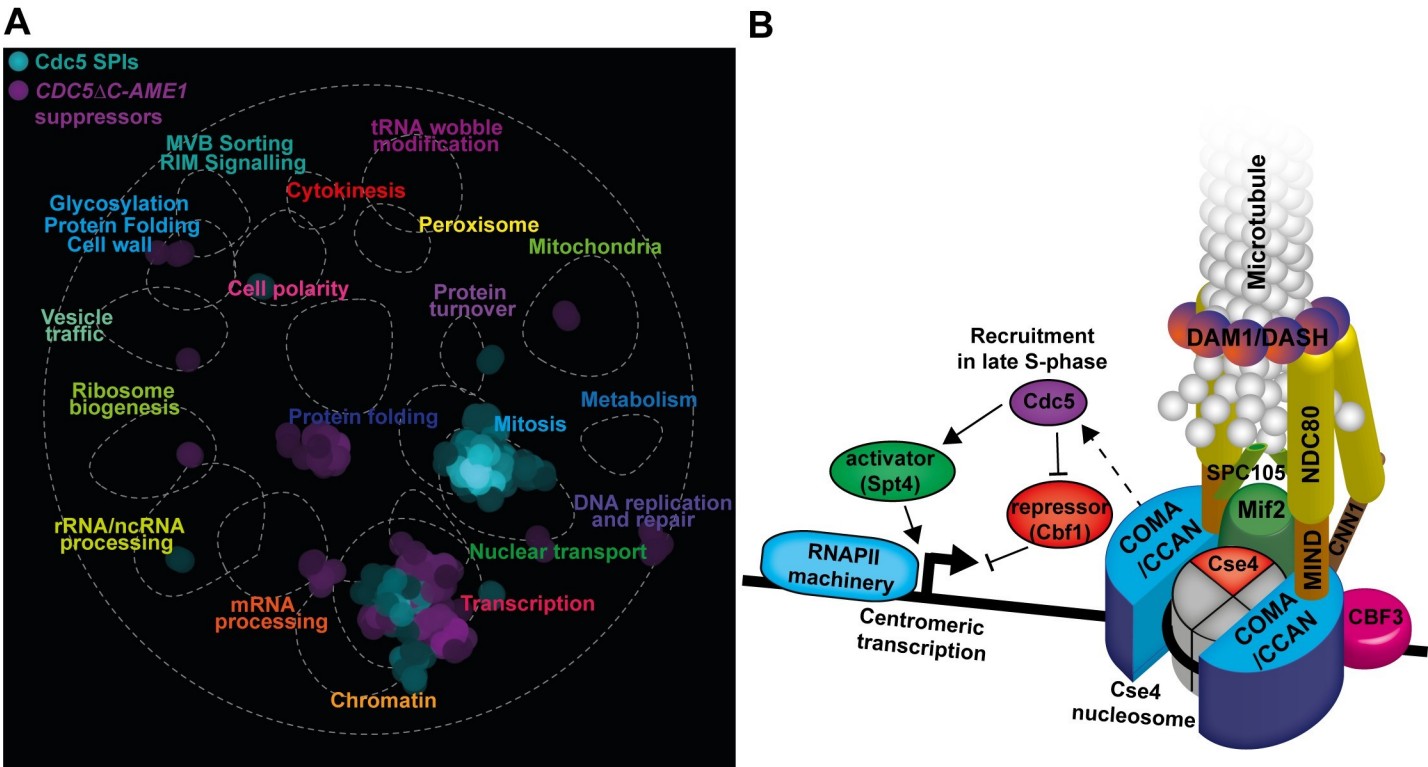

**Fig 7. Cdc5 recruitment to the CCAN influences centromeric transcription.** (A) SAFE analysis as in Fig 1B of the Cdc5 SPIs and the *CDC5ΔC-AME1* suppressor data identified highly dense regions that correspond to shared functions (mitosis and transcription for Cdc5 SPIs and transcription and mRNA processing for suppressors of *CDC5ΔC-AME1*). (B) A cartoon depicting Cdc5 recruitment to the kinetochore COMA complex (CCAN) resulting in increased production of CEN RNAs. We hypothesize that Cdc5 recruitment in late S-phase may influence CEN transcription by inhibiting a repressor such as Cbf1 and/or promote an activator such as Spt4 (see text for further details).

local centromeric chromatin architecture are important for kinetochore function in budding yeast and that Cdc5 kinetochore recruitment can perturb this.

## Does Cdc5 upregulate CEN RNA levels by inhibiting a repressor and/or promoting an activator of CEN transcription?

Based on both the literature and our findings, we suggest that Cdc5 plays a role in controlling the timing of cell-cycle specific CEN RNA elevation in budding yeast. Two potential mechanisms to regulate CEN RNA levels outside S-phase include repression by Cbf1 and control of Spt4/5 complex activity specifically at the centromere—either one could involve Cdc5 (Fig 7B).

It has been shown that Cbf1 can act as a transcription factor for specific genes transcribed by RNAPII [123], and until recently Cbf1 was thought to be required for CEN transcription [124]. However, new findings show that Cbf1 more likely functions in repressing CEN transcription [92,93]. Furthermore, Cbf1 can inhibit transcription along with Isw1 chromatin remodeler by displacing the TATA-binding protein (TBP) from promoters [125]. Notably, we did not find that *cbf1Δ* suppressed *CDC5ΔC-AME1*, but we found that a mutant of the budding yeast TBP, *SPT15* (*spt15-P65S*), did suppress the *CDC5ΔC-AME1* fusion (S4 Data), suggesting that Cbf1 may repress CEN transcription by inhibiting Spt15 at the centromere. Large-scale phosphoproteomics studies have shown that Cbf1 contains numerous sites phosphorylated by Mec1[ATR]/Tel1[ATM], Ctk1[CDK12] and Ssn3[CDK8] [126–128]. Cbf1 also contains ubiquitination

sites [129,130], thus suggesting it may be controlled by proteasome-dependent degradation. However, characterization of these modifications is lacking and Cbf1 has not been shown to be phosphorylated by Cdc5. We found in contrast to overexpression of Cbf1, tethering it to the COMA complex partially mitigated the *CDC5ΔC-AME1* growth phenotype (Fig 6E), in agreement with the notion that Cbf1 represses CEN transcription. Centromeric transcription is constrained to S-phase in a Cbf1-dependent manner [92,93]. However, the detailed mechanism of how Cbf1 activity is reduced during this period remains elusive. Recently, it was reported that Cbf1 centromeric localization was regulated in a cell-cycle dependent manner and that DNA helicase Pif1 and RNase H were important in this process [93,131]. Cdc5 is tightly regulated during the cell cycle and its expression begins in S-phase, although its presence at the centromere peaks during mitosis [26]. We did not assess CEN RNA levels at different points in the cell cycle, which will be interesting to explore in future studies. Our data, however, support the notion that upregulation of CEN transcripts, outside S-phase, causes mitotic defects [92]. Moreover, our data showing an increase in Cse4 signal in response to *CDC5ΔC-AME1* induction (S13 Fig) are consistent with Cse4 elevation at centromeres in *cbf1Δ* cells [93]. The PBD of Cdc5 has been shown to interact with Cbf1 [25]. Thus, we speculate that Cdc5 may be involved in inhibiting or displacing Cbf1 in a highly regulated manner, during this brief period in S-phase resulting in increased CEN RNA levels. However, since deletion of *CBF1* is viable and the additional Cbf1 recruitment only partially suppressed the *CDC5ΔC-AME1* phenotype, we suspect other players are also involved.

Deletion and mutation of *SPT4* rescued the growth defect caused by a constitutive Cdc5 recruitment specifically to the COMA complex. We showed this both in cells expressing the direct *CDC5ΔC-AME1* fusion (Fig 5B and 5C), which was also rescued by *spt4-138* mutant (S10E Fig), and also in Ame1-, Okp1- and Mcm21-GFP cells expressing *CDC5ΔC-GBP* (Fig 5E and S10A and S10B Fig), but not in other kinetochore-GFP cells (Fig 5D). These data indicate that the growth phenotype is caused by association of Cdc5 explicitly with the COMA complex rather than interfering with the N terminus of Ame1 or the kinetochore in general. We found that recruitment of Cdc5 to the COMA complex declustered outer kinetochore proteins (Fig 4 and S7 Fig). Is it possible that this is a consequence of hyperactivation of CEN transcription during mitosis? Using the GALCEN1-16 strain we showed that a perpetual CEN transcription leads to aberrant Mtw1-YFP foci (Fig 6D), however we note that this strain has to be grown on galactose for a relatively long time (>6 hours) before a clear phenotype can be observed. The clear scattered Mtw1-YFP phenotype we observe when *CDC5ΔC-AME1* is induced during metaphase could be a consequence of upregulating CEN RNA levels specifically during mitosis as opposed to persistent CEN transcription in the case of constitutively activating GALCEN1-16 on galactose. Alternatively, it is possible that the kinetochore destabilization observed by the forced Cdc5-Ame1 association causes the increased CEN RNA levels via an unrelated mechanism. However, this latter possibility is inconsistent with our findings showing that deletion of *SPT4* rescues kinetochore disruption phenotype in both *CDC5ΔC-AME1* expressing cells and in the GALCEN1-16 strain, indicating that reduction of CEN transcription or RNA levels rescues the aberrant kinetochores in both cases (Fig 6). Therefore, collectively our findings suggest that Spt4 is required for the declustered kinetochore phenotype caused by misregulation of CEN RNA levels outside S-phase.

Spt4 has an elusive kinetochore function in addition to its transcription elongation and mRNA-processing roles [76,77]. We identified a specific set of transcription elongation factors as *CDC5ΔC-AME1* suppressors which interact in the same pathway as Spt4 and are also involved in chromatin remodeling, including Spt5, Spt6, Spn1, Sgv1 and Ceg1 (Fig 5B and S9F–S9H Fig). For instance, the Sgv1/Bur1 kinase influences recruitment of the RSC complex [84] and was recently shown to phosphorylate a linker region of RNAPII to recruit histone

chaperone Spt6 [82]. Interestingly, the Spt6-RNAPII association is required for efficient recruitment of the Ccr4-Not mRNA-processing and deadenylation complex [132]. It was recently shown, in both Drosophila and human cells, that Spt6 is directly involved in CEN transcription and CENP-A maintenance [133]. Spt5 contributes to the recruitment of the mRNA capping enzyme [134]. Moreover, the Spt4/5 complex, Spt6 and the Sgv1 kinase influence the recruitment of the Paf1 histone modifier complex to active genes [78,135–137]. Intriguingly, the Paf1 complex facilitates the recruitment of the Set1/COMPASS complex which promotes methylation of the kinetochore DAM1/DASH complex [138]. Spt6 recruitment to elongating RNAPII depends on HDACs [139], some of which we identified as both kinetochore SPIs (S1C and S1D Fig) and *CDC5Δ-AME1* suppressors (S4 Data). Furthermore, in fission yeast, the Spt4/5 complex interacts with SWI/SNF-like chromatin remodeler Fun30$^{\text{S-}}$ $^{\text{MARCAD1}}$ which in collaboration with the FACT complex promotes elongation by RNAPII [140]. It has been reported that budding yeast Fun30 interacts with and supports centromere function and deleting it increases CEN transcription [141].

Based on our data, we are unable to discern whether Cdc5 is directly activating transcription at the centromere or indirectly stabilizing CEN RNA through an elusive mechanism such as centromeric chromatin reorganization or interactions with RNA. Nevertheless, there is precedent of both Cdc5 and Plk1 directly driving transcription of key mitotic regulators by phosphorylating specific transcription factors [142,143]. We note that the proteome-wide Cdc5 SPI screen identified transcription factors and components of the SAGA and RNAPII complexes (Table 1 and S2 Data). Together these findings provide the possibility that Cdc5 can drive the activation of CEN transcription, possibly through direct phosphorylation of subunits of the RNAPII machinery. This could also be achieved indirectly by the Cdc5 activity at the centromere/kinetochore, for example by the recruitment of transcription factors or histone modifiers or phosphorylation and activation of the SAGA complex that in turn facilitate transcription activation. In this scenario, we would expect that inhibition or deletion of these activators would suppress the *CDC5ΔC-AME1* phenotype and thus be found in our suppressor dataset. However, among the strongest suppressors of *CDC5ΔC-AME1* we did not find obvious candidates that are currently thought to be required for CEN transcription, such as known transcription factors, SAGA or FACT complex components. Nevertheless, we did find some components of the Mediator complex (S9F Fig) and TFIIH (S9H Fig) and a specific set of transcription elongation factors involved in histone modifications and mRNA processing. For instance, Spt4, Spt5, Sgv1, Spt6, Spn1, Npl16, Ldb7, Ceg1 and Hrp1 (Fig 5B and S9G, S9I, S9H Fig), which have genetic and physical interactions among each other, but also with a number of kinetochore proteins. These proteins are also involved in regulating histone modifications [144] and recruit chromatin modifiers, such as the FACT complex [145].

It has been shown that Spt4 is directly regulated by Hog1 kinase phosphorylation of threonine 42 and serine 43 (followed by proline 44) in response to osmotic stress [146], which provides an intriguing possibility that Cdc5 may also directly phosphorylate Spt4, specifically at the centromere, resulting in elevated CEN RNA. The PBD of Cdc5 interacts with Spt5, the binding partner of Spt4 [25], indicating that Cdc5 may bind Spt5 to target the phosphosites on Spt4. We found that forced recruitment of Cdc5 to the COMA complex resulted in increased Spt4- and Spt5-GFP signal (S10F and S10G Fig), suggesting that Cdc5 may be involved in stabilizing the Spt4/5 complex. Therefore, we would predict that a phosphatase involved in Spt4 dephosphorylation is important for controlling CEN RNA levels outside S-phase in a parallel mechanism to repression of CEN transcription by Cbf1. Phosphatases such as PP1 and PP2A act at the kinetochore to stabilize microtubule attachment by counteracting Aurora B and Mps1 kinase activity to silence the SAC [11–15,17]. The mechanism of these phosphatases at the kinetochore is well characterized, however another phosphatase with a role at the

kinetochore remains more elusive. Cdc14 phosphatase is the major phosphatase in budding yeast that reverses CDK activity during mitotic exit, preferring serine followed by proline as a substrate [147–149]. Cdc14 has been shown to localize at the kinetochore before anaphase onset, prior to its canonical release by the FEAR (Cdc14 early-anaphase release) and mitotic exit network, to dephosphorylate Dsn1 [150]. Interestingly, Cdc14 is involved in controlling Cdc5 localization at the SPB [151] and is also involved in repressing transcription in both human and yeast cells [152–154], thus making it the ideal candidate that could counteract Cdc5 in regulating CEN RNA levels by both controlling Cdc5 kinetochore localization and directly repressing transcription elongation, possibly by dephosphorylating Spt4. Of note, Cdc14 clustered with Cdc5 in our hierarchical clustering analysis of the SPI data (Fig 1C) and in our previous study, we found that forced recruitment of Cdc14 to kinetochore proteins, including COMA and MIND subunits, produced a mitotic phenotype, whereas a catalytically inactive mutant did not [18].

Based upon these data and our observations, we propose that a regulated Cdc5 activity at the kinetochore controls the cell-cycle specific timing of elevated CEN RNA levels. First, during late S-phase after CEN replication Cdc5 specifically localizes at the centromere at a low level via recruitment to the COMA complex to activate and stabilize Spt4/5 complex and in parallel inhibiting Cbf1 to increase CEN transcription. Second, prior to metaphase to anaphase transition, a phosphatase (such as Cdc14) reverses Cdc5 activity at the centromere/COMA complex and displaces it to other kinetochore substrates, possibly (but not limited to) to the outer kinetochore to influence microtubule attachment/dynamics. We speculate that this regulatory mechanism exists to control Cdc5 activity during mitosis to prevent persistently high CEN RNA levels.

## Material and methods

### Yeast methods and growth conditions

Unless stated otherwise yeast strains were grown at 30˚C in either yeast peptone dextrose medium (YPD; 1% yeast extract, 2% bacto-peptone and 2% glucose) or synthetic media containing 2% glucose and supplemented with amino acids and antibiotics depending on the plasmid/s being used. *S. cerevisiae* strains were constructed using standard yeast methods. Yeast CEN plasmids were created using the gap-repair cloning technique, which combines a linearized plasmid with PCR products using in vivo recombination. To create mutant variants of plasmids QuickChange Lightning Multi Site-Directed Mutagenesis Kit (Agilent Technologies) was used according to manufacturer's instructions. All PCR products were generated using primers from Sigma Life Science and Q5 polymerase (New England Biolabs, USA). Yeast strains and plasmids used in this study are listed in S1 Table.

### Yeast growth spot assay

Yeast cultures were grown overnight at 30˚C and cell density was measured and adjusted so that each culture had the same $OD_{600}$. Five- or ten-fold serial dilutions were performed and spotted onto appropriate solid media and incubated at 30˚C for 2–3 days unless otherwise stated.

### Synthetic physical interaction (SPI) and genetic suppressor assays

SPI screens were performed as previously described (Olafsson and Thorpe, 2015, 2018). For both the SPI and suppressor assays selective ploidy ablation (SPA; [108] was used to introduce plasmids into arrays of query yeast strains (GFP, deletion (Δ) and temperature-sensitive (ts)

mutant collections). Briefly, a Universal Donor Strain (W8164-2B), which contains conditional GALCEN centromeres, was transformed separately with the control and experimental plasmids. For example, for SPI screens either *pCUP1-CDC5-GBP* or *pCUP1-cdc5-kd-GBP* were used and for the suppressor screen *pGAL1-CDC5ΔC-AME1*. These universal donor strains were then mated with members of the GFP or Δ/ts collections arrayed with four or 16 replicates on 1536-colony rectangular agar plates using a pinning robot (ROTOR robot, Singer Instruments, UK). The resulting diploids were put through a series of sequential selection steps to maintain the query strain genome (GFP or Δ/ts mutations) and plasmid, while destabilizing and then removing the chromosomes of the universal donor strain by growing the cells in 5-FOA and galactose containing media (illustrated in S1A Fig). The resulting plates were scanned using a desktop flatbed scanner (Epson V750 Pro, Seiko Epson Corporation, Japan). Colony sizes were assessed and the resulting data analyzed using the ScreenMill suite of software [155]. The software calculates log growth ratios or Z-scores from the SPI data. For the kinetochore SPI screens, mean Z-scores or mean log growth ratios (LGRs) were calculated from two comparisons of colony sizes of GFP cells containing GBP-tagged kinetochore protein. The first comparison is with control colonies of the same GFP cells containing GBP alone. The second comparison is with control colonies with the relevant gene not tagged with GBP. These two controls account for the growth defects caused by, first, GBP-GFP binding and, second, ectopic expression of the gene. A positive Z-score or LGR indicates a growth defect compared to control colonies and a negative number can either indicate a growth enhancement compared to control colonies or poorer growth on control plates versus experiments. We note that some GFP-strains are sensitive to GBP alone which were excluded from the SPI analysis.

## Fluorescence microscopy

For live cell imaging yeast cultures were grown overnight at 23˚C in appropriate SC media (depending on plasmid being selected for) supplemented with 100mg/ml of adenine to minimize autofluorescence. Prior to imaging the cells were embedded in 0.7% low melting point agarose dissolved in growth medium. The cells were imaged with a Zeiss Axioimager Z2 microscope using a 63x 1.4NA oil immersion lens, illuminated with a Zeiss Colibri LED light source (CFP = 445nm, GFP = 470nm, YFP = 505nm, RFP = 590nm). Bright field contrast was enhanced using differential interference contrast (DIC) prisms. Images were captured using a Flash 4.0 LT CMOS camera with 6.5 μm pixels binned 2x2 (Hamamatsu photonics, Japan). Images were processed with ImageJ and Icy software.

## Fluorescence image analysis

Kinetochore foci intensity and sister kinetochore distance measurements were analyzed using a freely available semi-automatic ImageJ tool, FociQuant, which quantifies kinetochore foci fluorescence in a high-throughput manner and can be adapted to also measure distances between two foci [156]. In brief, the kinetochore foci are detected semi-automatically by first manually selecting the mitotic spindle region in budded cells, which contain separated sister kinetochores, the tool then uses the 'FindMaxima' function in ImageJ to automatically identify the sister kinetochore foci. In these experiments, the number of cells counted typically from a single experiment is indicated as n. The software fits a Gaussian plot to the intensity profile of each kinetochore focus in two dimensions (x and y) to accurately determine their position, allowing the distance between the two foci to be calculated. The FociQuant tools are available for download: https://sourceforge.net/projects/fociquantitation/files/.

## Cell cycle and spindle analysis

Yeast cells with fluorescently tagged tubulin (Turq2-Tub1), kinetochore (Dad4-YFP) and SPB (Spc42-RFP) carrying plasmids of interest, for example *pGAL1-CDC5ΔC-AME1*, were assessed using fluorescence microscopy. Overnight cultures growing in 2% raffinose 0.1% glucose SC–leu +ade media at 23˚C were resuspended in fresh 2% raffinose media and incubated until mid-log phase then resuspended in 2% galactose media and incubated for 4 hours before imaging. The cell-cycle stages were estimated based on bud, spindle and SPB morphology. Unbudded cells with a single kinetochore/SPB focus were categorized as G1-phase cells. Cells with small buds and single kinetochore/SPB focus were considered in S/G2-phase. Medium- or large-budded cells with two kinetochore/SPB foci in close proximity ($\leq$1.5μm and $\leq$3μm respectively) were considered metaphase cells. Cells with two kinetochore/SPB foci, one in the mother and the other in the daughter and connected by microtubules were classified as anaphase cells and telophase cells were scored as cells which had divided the kinetochore/SPB foci into the mother and daughter and spindle had disassembled. The number of cells counted from a single experiment are indicated as n.

The cell cycle status of yeast strains with fluorescently tagged kinetochore proteins such as Mtw1-YFP containing Cdc5-GBP-RFP or control plasmids was analyzed in a similar way. But since these cells did not have a fluorescently-tagged SPB or tubulin, we distinguished between cell-cycle stages by categorizing unbudded cells as G1, small-medium budded cells with a single kinetochore focus as S/G2, medium- and large-budded cells with two kinetochore foci as metaphase and large-budded cells with separated kinetochore foci in each daughter cell as anaphase/telophase.

To specifically assess metaphase-arrested cells, we engineered strains with CDC20 under the control of the repressible *MET3* promoter [60,61]. Since the *MET3* promoter is turned off by the addition of methionine, cells can be arrested in metaphase by adding methionine to the media to deplete Cdc20. We arrested the cells by growing them for two hours in media containing methionine, after which about 70% of cells were arrested as large-budded cells in metaphase. We also assessed cell-cycle progression after release from metaphase arrest by resuspending the cells in media lacking methionine and imaging cells after 2–3 hours of growth. The experimental setup can be visualized in S5A Fig.

## Bioinformatics analysis

For hierarchical clustering of the SPI data the Cluster (version 3.0) software [157] was used. We used hierarchical centroid linkage clustering of both the GBP screens and the GFP-tagged genes. The cluster diagram in Fig 1C was visualized using Java Treeview 1.1.6 [158]. Gene ontology enrichment analysis was performed using the GOrilla algorithm [159], available at: http://cbl-gorilla.cs.technion.ac.il/. Spatial Analysis of Functional Enrichment (SAFE) analysis [38] of the SPI data was performed using the Cell Map website (http://thecellmap.org).

## Flow cytometry

The protocol for sample preparation and cell cycle analysis using flow cytometry was adapted from [160,161]. Briefly, after growing cells to log phase ($OD_{600}$ = 0.6) they were fixed for 24 hours in 70% ethanol at -20˚C. The cells were collected by centrifugation and resuspended in 250μl of ribonuclease (RNase) solution (50mM Tris-HCl pH7.5, 100 g/ml RNaseA (Sigma-Aldrich)) and incubated at 37˚C for 3 hours. Cells were then washed once with water and resuspended in 500μl of pepsin solution (50mM HCl, 5mg/ml pepsin (Sigma-Aldrich)) and incubated at 37˚C for 1 hour. Next, the cells were collected by centrifugation at 13,000 rpm for 1 min and resuspended in 1ml SYTOX solution (50mM Tris-HCl pH7.5; 1μM SYTOX Green

nucleic acid stain (Invitrogen)) and incubated overnight at 4˚C. Finally, before FACS analysis the samples were sonicated for 10 sec using a microtip probe sonicator (Philip Harris Scientific). The cells were analyzed in a BD LSR II Flow cytometer. The resulting data were analyzed to calculate G1, S and G2/M populations using FlowJo 10.3 software.

## RNA extraction and RT-PCR

Total RNA from yeast cells was extracted using the RiboPureTM RNA Purification Yeast Kit (Life Technologies) according to manufacturer's instructions. RNA extracts were treated with DNase for one hour at 37˚C. To reverse transcribe the RNA 1st Strand cDNA Synthesis Kit for RT-PCR (Roche) was used according to manufacturer's instructions and 2000ng of RNA in a 20 µl reaction with random hexamers was used. PCR of the resulting cDNA was performed using DreamTaqTM Green PCR master mix (Thermo Scientific) with strand-specific primers against CEN1, 3 and 8 listed in S2 Table, together with primers against *CDC28* (as an internal control) in 50 µl PCR reactions. 35 PCR cycles were used.

## Supporting information

**S1 Table. Strains and plasmids used in this study.**
(PDF)

**S2 Table. Primers used for reverse transcription PCR.**
(PDF)

**S1 Fig. Kinetochore Synthetic Physical Interaction (SPI) screens.** (A) A schematic of the SPI method. A plasmid containing a gene of interest fused to GBP (e.g. *MTW1-GBP*) or control plasmids are introduced into a universal donor strain (UDS), containing a *URA3* locus and a conditional centromere on every chromosome. The UDS is mated with arrayed GFP strains, generating heterozygous diploid strains. For haploid SPI screens, the selective ploidy ablation technique is used to destabilize and select against the UDS chromosomes by growing the cells on media containing galactose and 5-FOA. For diploid SPI screens, the cells are kept as heterozygous diploids by growing them on synthetic glucose media. The screening is typically performed with 1536 colonies (96–384 strains with 4–16 replicates) arrayed on rectangular agar plates and a typical SPI screen takes a week to perform. (B) Venn diagram of SPI screens with GBP-tagged kinetochore proteins in haploid and heterozygous diploid GFP strains shows an overlap of 119 strain or ~50%. (C) The overlap of the 119 kinetochore SPIs found in inner and outer kinetochore SPI screens is shown. (D) Venn diagram showing outer kinetochore SPIs detected in both haploid and diploid GFP strains. Haploid-specific SPIs were excluded from this diagram and structural kinetochore proteins were also removed to highlight candidates of kinetochore regulation. Excluding the haploid-specific SPIs may omit interactions that affect kinetochore function; however, it also excludes growth effects caused by mislocalization of the GFP protein and so provides a conservative list of candidate kinetochore regulators. The Cnn1 is a subunit of the CCAN and thus should be technically considered an inner kinetochore protein, but it extends towards the outer kinetochore and many of the SPIs found in the Cnn1 screen overlap with outer kinetochore SPIs. ^ refers to GFP strains that were found as haploid and diploid SPIs with GBP-Cnn1 in contrast to Cnn1-GBP. Asterisk * refers to GFP strains that were also detected as haploid and diploid SPIs with Mtw1-GBP. Key for different colored protein names in (D) and (E) is shown below on the right. (E) Venn diagram as in (D) but showing inner kinetochore SPIs detected in both haploid and diploid GFP strains. The Chl4, Skp1 and Cbf1 SPI screens are not shown here since no SPIs were detected in diploid GFP

strains in those screens.
(TIF)

**S2 Fig. Cdc5-GBP constitutively colocalizes with GFP-tagged kinetochore proteins.** (A) Fluorescence microscopy with Ctf19-YFP (which binds GBP) and Mtw1-CFP (which does not bind GBP) to confirm that Cdc5-GBP and cdc5-kd-GBP are recruited to the kinetochore foci. (B) Examples of Cdc5-GBP recruitment to GFP-tagged kinetochore proteins. The resulting colonies from the SPI screen and the effect on growth indicated by log growth ratios (LGR) are shown on the right of the images for reference. All scale bars are 5μm. (C) Example of data from the Cdc5 kinetochore SPI screen showing each GFP strain arrayed with 16 replicates (in total 1536 colonies per plate). A cropped selection of GFP strains are shown on the right with Cdc5-GBP SPIs highlighted in red.
(TIF)

**S3 Fig. Associations of Cdc5 with kinetochore proteins produces a growth defect that is independent of *mad3Δ*.** (A) To assess the SAC, the Cdc5 SPI screen was repeated with a selection of GFP-tagged kinetochore strains in both wild-type and *mad3Δ* cells. Deletion of *MAD3* gene was not sufficient to suppress any Cdc5 kinetochore SPI except Cdc20-GFP. (B) Example of colonies from the Cdc5 kinetochore SPI screen with wild-type and *mad3Δ* GFP strains.
(TIF)

**S4 Fig. Cell-cycle analysis of the forced Cdc5-Dad4 interaction.** Asynchronous cultures of Dad4-YFP Turq2-Tub1 cells (T621) expressing *CDC5ΔC-GBP*, *cdc5ΔC-kd-GBP* or *GBP* alone, all under the control of GAL1 promoter were analyzed using fluorescence microscopy as in Fig 3C. Cells expressing *CDC5ΔC-GBP* (n = 144) are significantly increased in anaphase/telophase compared to *cdc5ΔC-kd-GBP* (n = 199) or *GBP* (n = 151) cells. Fishers exact test; p-values *** = $p < 10^{-3}$, **** = $p < 10^{-4}$. Error bars indicate 95% binomial C.I. The inset on the right shows a representative image of Dad4-YFP cells expressing Cdc5ΔC-GBP in anaphase. Scale bar is 5μm.
(TIF)

**S5 Fig. Analysis of the Cdc5-Mtw1 association phenotype.** (A) Diagram describing the experimental setup of the metaphase-arrest and release analysis. See text and methods for further details. (B) Mtw1-YFP Turq2-Tub1 cells were arrested in metaphase by incubation in media containing methionine (Cdc20 depletion). After two hours ~70% of cells were arrested in metaphase. Error bars indicate 95% binomial C.I. (C) After two hours of galactose induction of either *CDC5ΔC-GBP* or *cdc5ΔC-kd-GBP* in the metaphase-arrested cells the distance between two sister kinetochores was measured using a semi-automated quantification tool (see Materials and methods for details). The box and whiskers plot indicates the mean sister kinetochore distance and standard deviation of the variance (line and box, respectively). The whiskers indicate the 95 percentile and outliers are indicated as circles. Statistical analysis was done using two-tailed student's t-test; p-value *** = 5.4 x $10^{-8}$. Representative images are shown on the right. Scale bars are 5μm. (D) 10-fold serial dilutions spot assay with Mtw1-YFP (*mad1Δ*, *mad3Δ*, *bub1Δ* and *bub3Δ*) cells expressing either *CDC5ΔC-GBP* or *cdc5ΔC-kd-GBP* shows that the growth defect caused by the Cdc5-Mtw1 interaction is independent of the SAC. Interestingly expression of the *cdc5ΔC-kd-GBP* control became lethal in a *bub1Δ* strain.
(TIF)

**S6 Fig. Further analysis of the forced Cdc5-Ame1 association.** (A) 10-fold serial dilutions spot assay with *mad3Δ* strain expressing *CDC5ΔC-AME1*, *cdc5ΔC-kd-AME1* or *CDC5* alone shows that the growth defect caused by *CDC5ΔC-AME1* expression is not dependent on the

SAC. (B) Cell-cycle analysis by flow cytometry of cells expressing *CDC5ΔC-AME1*. Asynchronous log-phase cell cultures were grown in galactose for four hours before cell fixing and DNA staining with Sytox green and then measured with flow cytometry. The same G1 (red), S (green) and G2-M (yellow) gates were used for all samples. The colored numbers indicate the percentages of cells in each cell-cycle stage. This experiment was done in duplicate with identical results. See methods for further details. (C) Related to Fig 4F. After three hours of release from metaphase arrest, the majority of *CDC5ΔC-AME1* expressing cells displayed a severe mitotic spindle phenotype. Budded cells with two or more Mtw1-YFP foci from the analysis in Fig 4F were reanalyzed and cells with more than two foci, fractured or abnormal Mtw1-YFP signal were categorized as abnormal mitotic spindles. After both two (non-outlined bars) and three (black-outlined bars) hours *CDC5ΔC-AME1* expressing cells (n = 91 and n = 59) were significantly increased for abnormal kinetochores compared to *cdc5ΔC-kd-AME1* (n = 82 and n = 40) and empty plasmid control (n = 52 and n = 30). Fishers exact statistical test; p-values *** = $p < 10^{-3}$, **** = $p < 10^{-5}$. Error bars indicate 95% binomial C.I. Examples of cells expressing *CDC5ΔC-AME1* containing abnormal kinetochore foci are shown on the bottom. All scale bars are 5μm.
(TIF)

**S7 Fig. Cdc5-Ame1 association affects outer kinetochore proteins more than inner kinetochore proteins.** (A) Asynchronous cell cultures of Ndc80-GFP strain containing either *pGAL1-CDC5ΔC-AME1* or *pGAL1-cdc5ΔC-kd-AME1* plasmids were grown in galactose media for four hours before imaging with fluorescence microscopy. The Ndc80-GFP foci were analyzed and budded cells that exhibited declustered or scattered foci were quantified. Compared to *cdc5ΔC-kd-AME1* control (n = 107) cells expressing *CDC5ΔC-AME1* (n = 152) had significantly more cells with scattered Ndc80-GFP foci. Fishers exact test; p-values **** = $p < 10^{-5}$. Error bars indicate 95% binomial C.I. Representative images are shown on the right. All scale bars are 5μm. (B) The Ndc80-GFP foci intensities in (A) are shown as a box and whiskers plot. Cells expressing *CDC5ΔC-AME1* and *cdc5ΔC-kd-AME1* were compared but no significant difference was found between the two. The mean Ndc80-GFP intensity and standard deviation of the variance are indicated with a line and box, respectively. The whiskers indicate the 95 percentile and outliers are indicated as circles. Statistical analysis was done using two-tailed student's t-test. (C) Asynchronous cell cultures of Okp1-GFP strain containing either *pGAL1-CDC5ΔC-AME1* or *pGAL1-cdc5ΔC-kd-AME1* plasmids were analyzed in the same way as (A). Cells expressing *CDC5ΔC-AME1* (n = 273) exhibited slightly more cells with scattered Okp1-GFP foci compared to *cdc5ΔC-kd-AME1* expressing cells (n = 318). Fishers exact test; p-values * = $p < 0.05$. Representative images are shown on the right. (D) The Okp1-GFP foci intensities in (C) were compared between cells expressing *CDC5ΔC-AME1* and *cdc5ΔC-kd-AME1* as in (B). Cells expressing *CDC5ΔC-AME1* had a slightly lower Okp1-GFP foci intensity compared to *cdc5ΔC-kd-AME1* expressing cells. Statistical analysis was done using two-tailed student's t-test; * = $p < 10^{-4}$. (E) Asynchronous cell cultures of Ctf19-GFP strain containing either *pGAL1-CDC5ΔC-AME1* (n = 167) or *pGAL1-cdc5ΔC-kd-AME1* (n = 148) plasmids were analyzed in the same way as (A) and (C). There was no statistical difference (Fishers exact test) in scattered Ctf19-GFP foci between cells expressing *CDC5ΔC-AME1* and *cdc5ΔC-kd-AME1*. Representative images are shown on the right. (F) The Ctf19-GFP foci intensities in (E) were compared between cells expressing *CDC5ΔC-AME1* and *cdc5ΔC-kd-AME1* as in (B) and (D). There was no statistical difference (two-tailed student's t-test) in Ctf19-GFP intensity between cells expressing *CDC5ΔC-AME1* and *cdc5ΔC-kd-AME1*. (G) Asynchronous cell cultures of Mif2-GFP strain containing either *pGAL1-CDC5ΔC-AME1* (n = 236) or *pGAL1-cdc5ΔC-kd-AME1* (n = 153) plasmids were analyzed in the same way as (A) and (C). There was no

statistical difference (Fishers exact test) in scattered Mif2-GFP foci between cells expressing *CDC5ΔC-AME1* and *cdc5ΔC-kd-AME1*. Representative images are shown on the right. (H) The Mif2-GFP foci intensities in (G) were compared between cells expressing *CDC5ΔC-AME1* and *cdc5ΔC-kd-AME1* as in (B), (D) and (F). There was no statistical difference (two-tailed student's t-test) in Mif2-GFP intensity between cells expressing *CDC5ΔC-AME1* and *cdc5ΔC-kd-AME1*.

(TIF)

**S8 Fig. The growth phenotype caused by the Cdc5-Ame1 association is not dependent on hyperphosphorylation of Ame1 or Cse4.** (A) The amino acid sequence of the N terminus of Ame1 and homologs of several related yeast species. Reported phosphorylation sites of Ame1 are shown in bold (thebiogrid.org). ClustalW alignment (www.ebi.ac.uk/Tools/msa/clustalo/) shows the evolutionary conservation of Ame1 N terminus among several yeast species. *S. bay* = *Saccharomyces bayanus*, *S. mik* = *Saccharomyces mikatae*, *S. cer* = *Saccharomyces cerevisiae*, *S. par* = *Saccharomyces paradoxus*. The seven phosphoserines within N-terminal Ame1 are highlighted green. (B) 10-fold serial dilutions spot assay with cells expressing *ame1-7A* control or *CDC5ΔC-ame1-7A* fusion (serines 41, 45, 52, 53, 59, 96 and 101 mutated to alanines) suggests that the growth defect caused by the Cdc5-Ame1 association is not dependent on N-terminal Ame1 phosphorylation. (C) 10-fold serial dilutions spot assay with *cse4-9A* strain expressing *CDC5ΔC-AME1*, *cdc5ΔC-kd-AME1* or empty plasmid control suggests that the growth defect caused by *CDC5ΔC-AME1* expression is not dependent on N-terminal Cse4 phosphorylation.

(TIF)

**S9 Fig. Data from the *CDC5ΔC-AME1* suppressor screen.** (A) Venn diagram showing the overlap between gene deletion (Δ) suppressors of *CDC5ΔC-AME1* growth phenotype and suppressors of *CDC5* overexpression. (B) Examples of the strength of suppression of the growth defects. Colonies that were four times as big as the plate median colony size was used as a cutoff for suppression. We estimate that as a conservative threshold since some plates had a higher frequency of larger colonies. We organize the data according to the strength of suppression; + weak, ++ moderate and +++ strong suppression compared to wild type. See S4 Data for all suppressor data and methods for further details. (C) Venn diagram showing the overlap of temperature-sensitive mutant (ts) suppressors of *CDC5ΔC-AME1* growth phenotype at three different temperatures (23°C, 26.5°C and 30°C). In total 43 suppressors. (D) Venn diagram showing the overlap of temperature-sensitive mutant (ts) suppressors of *CDC5ΔC-AME1* growth phenotype (43 in total) and *CDC5* overexpression (13 in total). (E-H) A selection of cropped images of colonies from the suppressor screens. Mutants or deletions of components of the Mediator complex (F), the RSC complex (G), TFIIH (H), Spn1/Iws1 (I), the Condensin complex (J), and mRNA-processing (H) were found as suppressors of the growth defect caused by *CDC5ΔC-AME1*.

(TIF)

**S10 Fig. A functional Spt4/5 complex is required for the growth defect caused by Cdc5-COMA associations.** (A-B) 10-fold serial dilutions spot assay with wild-type and *spt4Δ* Okp1-GFP (A) and Mcm21-GFP (B) strains expressing *CDC5ΔC-GBP* or *GBP* control shows that the growth defect caused by the Cdc5-COMA associations depends on *SPT4*. (C) Analysis of Ame1-GFP cells, which were imaged with fluorescence microscopy after 4 hours of growth in 2% galactose media to induce expression of either *pGAL1-CDC5ΔC-GBP* (n = 566) or *pGAL1-GBP* (n = 718). Cells that did not show RFP/GFP colocalization were excluded from this analysis. Cells expressing Cdc5ΔC-GBP-RFP were significantly increased in metaphase

compared to control. Fishers exact statistical test; p-values **** = p = 3.38 x $10^{-11}$. Error bars indicate 95% binomial C.I. Representative images are shown on the right. Scale bars are 5μm. (D) Ame1-GFP foci in cells expressing *cdc5ΔC-kd-AME1* or *CDC5ΔC-AME1* were analyzed with fluorescence microscopy and the data is shown as box and whiskers plots. The mean Ame1-GFP intensity and standard deviation of the variance are indicated with a line and box, respectively. The whiskers indicate the 95 percentile and outliers are indicated as circles. Asynchronous cell cultures of Ame1-GFP strain containing either an empty, *pGAL1-cdc5ΔC-kd-AME1* or *pGAL1-CDC5ΔC-AME1* plasmids were grown in galactose media for four hours before imaging. Compared to the empty plasmid control (n = 96), expression of either *cdc5ΔC-kd-AME1* (n = 31) or *CDC5ΔC-AME1* (n = 79) had significantly reduced Ame1-GFP signal. Two-tailed student's t-test; p-values **** = p $< 10^{-10}$. Representative images are shown on the right. Scale bars are 5μm. (E) 10-fold serial dilutions spot assay showing that a *spt4-138* mutant can also rescue the growth defect caused by *CDC5ΔC-AME1*. (F-G) Spt4-GFP and Spt5-GFP fluorescence signals were analyzed after inducing either *CDC5ΔC-AME1* or *cdc5ΔC-kd-AME1* for four hours as in (D). Statistical analysis was done using two-tailed student's t-test; **** = p $<10^{-8}$, *** = p $<10^{-4}$. Representative images are shown on the bottom. Scale bars are 5μm.
(TIF)

**S11 Fig. *SPT4* is not required to drive expression by the *GAL1* promoter.** (A) 10-fold serial dilutions spot assay with wild-type and *spt4Δ* strains overexpressing *PSH1* or empty control (top panel) and a 5-fold serial dilutions spot assay with wild-type and *spt4Δ* strains overexpressing *BIM1*, *BIK1* or empty control (bottom panel). Deletion of *SPT4* does not suppress the growth defect caused by the overexpression of any of the genes. (B) 10-fold serial dilutions spot assay with wild-type and *spt4Δ* strains expressing either *CDC5ΔC-AME* or empty control and coexpressing *SPT4* driven by either a *CUP1* promoter (top 4 rows) or *GAL1* promoter (bottom 4 rows). In both cases the coexpression of *SPT4* represses the *spt4Δ*-dependent rescue of the growth defect caused by *CDC5ΔC-AME1*.
(TIF)

**S12 Fig. Inhibition of transcription elongation is not sufficient to suppress the growth defect caused by Cdc5-Ame1.** (A) 5-fold serial dilutions spot assay with wild-type cells expressing either *CDC5ΔC-AME1*, *cdc5ΔC-kd-AME1* or empty plasmid. (B) Spot assay as in (A) was repeated with two concentrations of 6-Azauracil (6-AU) added to the media. The agar plates were incubated for two additional days (4 in total) at 30˚C. (C) Spot assay as in (A) was repeated with two concentrations of mycophenolic acid (MPA) added to the media. The agar plates were incubated for two additional days (4 in total) at 30˚C.
(TIF)

**S13 Fig. Cse4 analysis in cells expressing the Cdc5-Ame1 association.** (A) Asynchronous cell cultures of Cse4-YFP (internally tagged) strain containing either *pGAL1-CDC5ΔC-AME1* or *pGAL1-cdc5ΔC-kd-AME1* plasmids were grown in galactose media for four hours before imaging with fluorescence microscopy. The Cse4-GFP foci intensity was quantified. Compared to *cdc5ΔC-kd-AME1* control (n = 456) cells expressing *CDC5ΔC-AME1* (n = 483) had significantly increased Cse4-YFP foci intensity. The mean Cse4-YFP intensity and standard deviation of the variance are indicated with a line and box, respectively. The whiskers indicate the 95 percentile and outliers are indicated as circles. Statistical analysis was done using two-tailed student's t-test; **** = p $<10^{-8}$. Representative images are shown on the right. Scale bars are 5μm. (B) Cse4-YFP (internally tagged) cells were arrested in metaphase using Cdc20 depletion and large-budded cells containing a single Cse4-YFP focus or two foci were quantified.

Cells expressing *CDC5ΔC-AME1* (n = 222) during metaphase-arrest had increased number of cells with a single Cse4-YFP focus compared to cells expressing *cdc5ΔC-kd-AME1* (n = 161). Fishers exact test; p-values **** = $p < 10^{-10}$. Error bars indicate 95% binomial C.I. (C) The Cse4-YFP signal from cells in (B) was measured along the mitotic spindle after inducing *CDC5ΔC-AME1* or controls for two hours (see S5A Fig for further description of experimental setup). A 4 µm line with 19 points was used to include background signal and to cover the spread signal phenotype in cells expressing *CDC5ΔC-AME1* (40 randomly selected mitotic spindles were measured for each condition). The shadowed area indicates standard deviation. Representative images are shown on the right. Scale bars are 5µm.
(TIF)

**S14 Fig. Additional RT-PCR experiments and analysis of *spt4Δ* cells expressing the forced Cdc5-Ame1 interaction.** (A) Reverse transcription PCR (RT-PCR) experiments were repeated as in Fig 6A. Quantification of the relative CEN RNA levels is shown in the graph on the right. Ratio of the *CDC28* RNA control to the *CEN* RNAs was calculated. (B) Cells from the analysis in Fig 6B were released into anaphase and analyzed after two (non-outlined bars) and three (black-outlined bars) hours as in S6C Fig. The abnormal kinetochore phenotype seen in Fig 4F and S6C Fig was reduced by *spt4Δ* and there was no significant difference between *spt4Δ* cells expressing *CDC5ΔC-AME1* (n = 127 and n = 132), cdc5ΔC-kd-AME1 (n = 141 and n = 150) or empty plasmid controls (n = 137 and n = 132). Fishers exact statistical test. Error bars indicate 95% binomial C.I.
(TIF)

**S15 Fig. Overexpression of *CBF1* is not sufficient to suppress the Cdc5-Ame1 association growth phenotype.** (A) 5-fold serial dilutions spot assay with cells expressing either *CDC5ΔC-AME* or *cdc5ΔC-kd-AME1* and coexpressing either a pCUP1-driven *CBF1* or an empty plasmid control with either no copper (CuSO₄) added to the media for low expression of *CBF1*, or for increased *CBF1* expression 50µM or 200µM was added. In none of the cases was overexpression sufficient to suppress the growth defect caused by *CDC5ΔC-AME* expression. (B) 5-fold serial dilutions spot assay with cells expressing either *CDC5ΔC-AME* or *cdc5ΔC-kd-AME1* and either overexpressing *CBF1* using the *GAL1* promoter or an empty plasmid control. Overexpression of *CBF1* driven by pGAL1 was insufficient to suppress the *CDC5ΔC-AME* growth phenotype.
(TIF)

**S1 Data. Proteome-wide Ctf19 and Cnn1 synthetic physical interaction data.** The Z-scores are shown for individual synthetic physical interaction using either gene (pHT311 for Ctf19 and pHT318 for Cnn1) or GBP (pHT4) control for each GFP strain in four replicates. A mean Z-score value from the two controls are highlighted in bold. The SPI screens were performed both in haploid and heterozygous diploid cells and the data for individual screen is separated in four tabs.
(XLSX)

**S2 Data. Kinetochore-specific synthetic physical interaction data.** SPI screens with 12 kinetochore GBP fusions were performed with 439 GFP strains, each with at least four replicates. The mean log growth ratios are shown for individual synthetic physical interaction using the gene and GBP controls for each GFP strain. The screens were repeated in heterozygous diploid strains and all haploid (hap) and diploid (dip) data are shown in the same tab for easier comparison.
(XLSX)

**S3 Data. Cdc5 synthetic physical interaction data.** The Z-scores or log growth ratios (LGRs) are shown for individual synthetic physical interaction of Cdc5-GBP compared with three controls; *CDC5* alone (pHT426), kinase-dead Cdc5-GBP (cdc5-T242A-GBP; pHT444) and a kinase-dead PBD mutant (cdc5-kd-PBD*-GBP; pHT503) for each GFP strain (with four or 16 replicates) and the mean Z-score value from the three controls are highlighted in bold. The individual SPI screens are separated in three tabs: First, the proteome-wide Cdc5 SPIs. Second, the kinetochore Cdc5 SPIs using the GBP control. Third, the kinetochore Cdc5 SPIs using the cdc5ΔC-kd-GBP controls.
(XLSX)

**S4 Data. *CDC5ΔC-AME1* and *CDC5* suppressor data.** Mean growth ratios, from four replicates, of *pGAL1-CDC5ΔC-AME1* (pHT568) and *pGAL1-CDC5* (pHT573) compared to pGAL1-empty (pGAL1-empty) are shown for each deletion and temperature-sensitive (ts) mutant. The colony sizes are normalized to the plate median colony size, which results in a conservative estimate of suppressors. The strength of suppression is organized by growth ratio of 4–6 as weak (+), 7–9 as moderate (++) and 10> as strong (+++). Growth ratios of <4 were not considered as suppressors. The individual suppressor data (deletion strains and ts mutants at 23˚C, 26.5˚C and 30˚C) is split between tabs.
(XLSX)

## Acknowledgments

We thank F Uhlmann, CA Müller, J Diffley, R Reid, J Svejstrup, A Musacchio, S Biggins, S Hedouin, GJ Kops, AT Saurin, M Basrai, P Mishra, I Cheeseman, F Caudron and V Draviam for discussions and suggestions. We are grateful to CA Müller and M Basrai for strains and D Peer, H Caulston, G Brown, B Andrews, R Rothstein, J Dittmar and members of the Thorpe lab for other help. We thank D Conti and RSM Howell for helpful comments on the manuscript.

## Author Contributions

**Conceptualization:** Guðjón Ólafsson, Peter H. Thorpe.

**Formal analysis:** Guðjón Ólafsson.

**Funding acquisition:** Peter H. Thorpe.

**Investigation:** Guðjón Ólafsson, Peter H. Thorpe.

**Methodology:** Guðjón Ólafsson.

**Project administration:** Peter H. Thorpe.

**Resources:** Peter H. Thorpe.

**Software:** Guðjón Ólafsson.

**Supervision:** Peter H. Thorpe.

**Writing – original draft:** Guðjón Ólafsson, Peter H. Thorpe.

**Writing – review & editing:** Guðjón Ólafsson, Peter H. Thorpe.

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
