## [Decision Letter · Decision Letter 0]

15 Jun 2020

Dear Dr Thorpe,

Thank you very much for submitting your Research Article entitled 'Polo kinase recruitment via the constitutive centromere-associated network at the kinetochore elevates centromeric RNA.' to PLOS Genetics. Your manuscript was fully evaluated at the editorial level and by three independent peer reviewers. The reviewers all agreed that the study was interesting, focused an important topic, and provided important new data to the field, however, they identified a few aspects of the manuscript that should be improved.

We therefore ask you to modify the manuscript according to the review recommendations before we can consider your manuscript for acceptance. Your revisions should address the specific points made by each reviewer. We understand that it might not be feasible to address all comments experimentally, but encourage the incorporation of any additional existing data that addresses the reviewers' concerns. The revision should also include:

a. clarification or simplification of  the SPI screening approach to educate and aid readers who may be less familiar with this technique (as noted by all reviewers)

b. incorporation of error bars, statistical measures, for multiple figures (as noted by several reviewers)

c. clearer rationale for the focus on Cdc5 considering the other hits/interactions that were revealed by the screen

d. additional explanation or elaboration on the growth arrest phenotype

e. insight into the relationship between Ame1 and Cdc5 (see R2 comments)

[LINK]

Yours sincerely,

Beth A. Sullivan, PhD

Associate Editor

PLOS Genetics

Gregory P. Copenhaver

Editor-in-Chief

PLOS Genetics

Reviewer's Responses to Questions

**Comments to the Authors:**

Reviewer #1: This manuscripts employs Synthetic Physical Interactions (SPIs) screens, mutagenesis and growth assays in yeast to investigate the interactions between kinases and phosphatases and the kinetochore focusing in particular on the Polo kinase/Cdc5. The ‘systems’ approach taken by this work, allows the authors to build on known and novel interactions to shed light into the complex regulation of kinetochore proteins by phosphorylation. Importantly, this work reveals a possible role for Polo kinase in regulating CEN transcription.

By forcing the recruitment of Cdc5 to different kinetochore subcomplexes, the authors identify multiple effects that are consistent with the many roles that Polo kinase plays at the kinetochore. Interestingly, phenotypes resulting from the constitutive recruitment of Cdc5 to the COMA complex are rescued by mutations in genes involved in chromatin remodeling and transcriptional elongation. One of these phenotypes is an increase in CEN RNA levels and this phenotype is dependent on the transcriptional regulator Spt4. Recruitment of proteins involved in transcription to the kinetochore also cause caused growth phenotypes, suggesting that perturbing normal CEN RNA level is deleterious for cell cycle progression.

Cdc5 could be influencing CEN transcription either directly or indirectly and, given the SPI interactions between Cdc5 and transcriptional regulators, the authors speculate that the role of Cdc5 may be direct, such as via the phosphorylation of RNAP II.

I only have a couple suggestions below.

1) I struggled connecting figure 1 (five SPIs for kinetochore proteins three of which were previously published) with the rest of the study, which focuses on Cdc5. Making this connection clearer would help the reader.

2) I’m not familiar with the SPI system, could the authors include a conceptual summary of the approach when introducing the screen in Figure 1A, so the reader does not need to read previous papers to understand what’s being done here? The illustration is clear in terms of what the approach consists of, but clearly stating what biological information is gained by forcing proteins to come together would be helpful. What is the meaning of a slower growth when 2 proteins are forced together? Being such an artificial system, its advantage is not immediately apparent. It’s clearly an established and valid approach used routinely in other papers from this lab, it just needs a little bit of explanation.

Reviewer #2: Title: Polo kinase recruitment via the constitutive centromere-associated network at the

kinetochore elevates centromeric RNA.

Authors: Guðjón Ólafsson and Peter H. Thorpe

Understanding the regulation of kinetochore function is an important area of research because of its significance to cancer biology. Despite this, we have limited understanding of structural and functional architecture of the kinetochores. In this manuscript, authors have used high-through put synthetic protein interaction (SPI) approach to identify genes involved in the regulation of kinetochore function in budding yeast. In this study, authors performed two SPI screens and compared these results with the data from their three previously published screens. The SPI screens showed enrichment of proteins involved largely in post-translational modifications for example, protein kinases, phosphatases and chromatin modifiers. They performed in-depth investigation on effects of constitutive Cdc5 kinase activity and identified kinetochore as a major target where targeting Cdc5 to different kinetochore components resulted in phenotypes that were consistent with the role of different subcomplexes of the kinetochore. Authors suggest that constitutive Cdc5 kinase activity at kinetochore is mediated by centromeric RNA via Spt4-regulated process.

The manuscript is very well-written, nicely organized into sections, however, it requires simplification for the broad audience of the journal. The objectives of this study are logical, broad, hypothesis-driven, and thought-provoking. The results presented are experimentally sound, constitute novel addition to the literature in the field, and provide deeper research insights into the current state of knowledge in centromere biology and chromosome segregation.

Specific comments are below:

1) The authors should give a bit more introduction to SPI as this is the crux of the whole paper.

2) Authors have utilized a fusion construct of Cdc5 and Ame1 to examine the effect of constitutive association of Cdc5 with kinetochores. However, their previous studies have used independent constructs (Cdc5-GBP and kinetochore proteins tagged with GFP), which were forced to interact in-vivo upon their induction. The caveat in the fusion construct is that it does not allow to differentiate the contribution of individual proteins to the phenotypes observed in this study. Is it unclear why authors have not tried tagging the N-terminus of Ame1 with the GFP as they have done for other kinetochore proteins in their previous studies? Does Ame1-GFP and Cdc5-GBP exhibit SPI phenotype.

3) It is unclear whether overexpression of Ame1 can compete out the cdc5∆C-Ame1 and suppress its associated growth defect? It would be useful to know if cdc5∆C kd-Ame1 can complement Ame1 null. Is the essential function of Ame1 intact in this construct? Similarly, I am wondering if Cdc5-Ame1 can complement a cdc5 mutant phenotype when it is not overexpressed.

4) Authors have reported severe kinetochore and spindle defects phenotype that they have contributed to constitutive Cdc5-Ame1 association. However, these phenotypes are hallmarks of defective Dynein function. Ame1 is known to exhibit negative genetic interaction with Dynein (please see, SGD); and fusing Cdc5 with Ame1 may potentially affect its interaction with Dynein components at the kinetochores. In line with this explanation, authors have also observed Dynein components in their list of Cdc5 SPI proteins. It is therefore unclear whether phenotypes observed in this study are contributed solely due to Cdc5-Ame1 fusion or also mediated by defective Dynein function. What are their thoughts on this? Separation of function alleles could be used for future studies.

5) Authors have proposed a role for Cdc5 in regulation of CEN transcription outside of the S-phase, which includes Cbf1 and/or Spt4. If this conclusion holds true, then conditional depletion of Cdc5 may allow the accumulation of CEN transcripts. It would be interesting to examine the CEN transcript levels outside of the S-phase after depletion of Cdc5 (perhaps GAL-Cdc5).

6) Authors have concluded that the growth arrest phenotype caused by forced association of Cdc5 with kinetochore is independent of the SAC. Similar conclusions have been drawn for Cdc5∆C data. However, authors have not examined mad2∆ which has documented evidence of genetic interaction with Cdc5. To better understand the role of SAC, examination of phenotypes in combination with mad2∆ will be helpful. I don’t think that the results will change but I am curious.

7) For suppressor screen with cdc5∆C-Ame1, the authors used the Cdc5 as a control for lethality. To enrich the suppressors that are specific for Ame1-dependent forced recruitment of Cdc5, I thought it would have been worthwhile to perform the suppressor screen with Cdc5-Ame1 (with intact Polo box) and still use Cdc5 as a control or filter.

8) Line 18 (abstract): “where”? please correct.

9) Line 118: “six additional kinetochore proteins fused to GBP”. Why only these six were selected? Why not other kinetochore proteins from the Table 1? Please provide explanation for this selection.

10) Lines 120-122: Authors tagged proteins with -GBP and -GFP. It is important to determine whether tagging may contribute to the observed SPI. Do tagging of these proteins affect their structure-function, in other words, are there any phenotypes caused by these tags?

11) Are most of the binary SPI identified with Cdc5 reproducible when reciprocal fusion with GFP and GBP between Cdc5 and targeted kinetochore protein were tested? One would expect that swapping the fusion will more likely not affect the outcome.

12) Lines 128-129: Why only Cdc5 was selected for further study? It seems like it was based on their previous study with Cdc5, but additional explanations and clarifications would be helpful.

13) Lines 195-196: “ what was striking form these data was the number of kinetochore proteins represented in this set of Cdc5”. Please specify how many kinetochore proteins?

14) Lines 300-318: Cell cycle arrest in metaphase observed upon forced Cdc5-Ame1 association could be due to fusion construct, hence this observation is different from previous results for MIND, Cse4 and Dam1/Dash complex. Also, it is unclear why cells accumulate with G1 DNA content despite their metaphase spindle phenotype. Is it possible that cells are stuck in late telophase with aberrant spindles due to issues with cytokinesis. Please explain.

15) Line 349: Authors stated that SPI phenotype was minimal for Okp1-GFP and absent for Ctf19-GFP. Curious why Mcm21-GFP was not examined and if it was done, then what phenotype were observed?

16) Line 438: Authors have used spt4-138 allele to suppress the Cdc5delC-Ame1 phenotype. What is the status of CEN transcription and whether it is increased in this mutant allele? Also the transcription phenotype was observed for CEN3 and CEN1 but not CEN8. Any explanation for this difference?

17) Figure 1C, the term “Log Growth Ratio” is a little confusing, since, based on the color (yellow to blue), it should be “Log growth inhibition ratio”.

18) Figure 6A: Although the data for CEN transcription are thoroughly evaluated and appropriately analyzed, it is unclear if this experiment was repeated to confirm the findings.

19) Grammatical and spelling errors in abstract (lines 13-14 and 18, respectively)

20) Scale bar missing for Figure 3B and 3D.

Reviewer #3: Error-free chromosome segregation relies on the ability of kinetochores to make correct attachments to the spindle microtubules. This fundamentally critical role of kinetochore is tightly controlled by numerous protein-protein interactions often modulated by regulators such as kinases and phosphatases. We have a pretty good understanding of the kinetochore composition and good knowledge on the identities of several kinetochore regulators. However, most kinetochore regulators possess diverse roles involving multiple kinetochore components and as a consequence, dissecting and understanding specific roles of essential regulators often prove difficult. Here, Olafsson and Thorpe employ a Synthetic Protein Interaction (SPI) screen, which was successfully used by the authors in the past to gain insights into the role of Cdc14 and checkpoint activation in budding yeast, to unravel novel roles of Cdc5 (Polo-like kinase) at the kinetochore. They show that forced targeting of Cdc5 to distinct kinetochore subcomplexes resulted in growth phenotype in a kinase dependent manner and the phenotypes vary depending on when and where the Cdc5 was recruited. Particularly, the forced association of Cdc5 with the Constitutive Centromere Associated Network (CCAN) component Ame1 (CENP-U) resulted in kinetochore delustering and impaired metaphase to anaphase transition. Finally, their work also shows that the CCAN recruitment of Cdc5 increased centromeric RNA via Spt4/5 complex, a complex involved transcription elongation and mRNA processing.

In my opinion, this manuscript reports a lot of interesting observations. I do note, that with the employed approach it is hard to confidently rule out possible indirect consequence of constitutive targeting of Cdc5. Having said that, I feel that the authors do a good job of critically analysing their observations in the light of published literature. Overall, I feel that this work will greatly benefit the mitosis community in general and kinetochore field in particular. All my concerns are minor and are outlined below:

• Point 3 line 164 – Not clear how it was decided that cdc5 is important when it is only seen in 3/12 screens using these cut off criteria? Why not cdc28?

• Are cdc5 experiments done in haploid and diploid?

• Line 356: Might be useful to include a panel with the N-terminal sequence of Ame1 and the Cdc5 consensus sites to appreciate with sites are mutated.

• Fig 3, 4E, 4F What size are the scale bars in the yeast images? No scale bars in S10D pictures, but ref in legend.

• Figure 4C, please include details of the error bars in figure legend

• Line 278 – there is no S5E, should be S5D

• Fig S5C: how many KTs were quantified? What do the circles represent?

• Fig 6A (right panel): no error bars?

• Line 529: representative images of Fig 6B would help?

**Have all data underlying the figures and results presented in the manuscript been provided?**

Reviewer #1: Yes

Reviewer #2: Yes

Reviewer #3: Yes

PLOS authors have the option to publish the peer review history of their article (what does this mean?). If published, this will include your full peer review and any attached files.

Reviewer #1: No

Reviewer #2: No

Reviewer #3: No

---

## [Editor Report · Decision Letter 1]

13 Jul 2020

Dear Dr Thorpe,

Thank you for submitting your revised manuscript and for carefully addressing the reviewers' and editor's comments. We are pleased to inform you that your manuscript entitled "Polo kinase recruitment via the constitutive centromere-associated network at the kinetochore elevates centromeric RNA." has been editorially accepted for publication in PLOS Genetics. Congratulations!

Yours sincerely,

Beth A. Sullivan, PhD

Associate Editor

PLOS Genetics

Gregory P. Copenhaver

Editor-in-Chief

PLOS Genetics

Comments from the reviewers (if applicable):

**Data Deposition**

http://datadryad.org/submit?journalID=pgenetics&manu=PGENETICS-D-20-00683R1

**Press Queries**

---

## [Editor Report · Acceptance letter]

12 Aug 2020

PGENETICS-D-20-00683R1 

Polo kinase recruitment via the constitutive centromere-associated network at the kinetochore elevates centromeric RNA 

Dear Dr Thorpe, 

We are pleased to inform you that your manuscript entitled "Polo kinase recruitment via the constitutive centromere-associated network at the kinetochore elevates centromeric RNA" has been formally accepted for publication in PLOS Genetics! Your manuscript is now with our production department and you will be notified of the publication date in due course.

With kind regards,

Matt Lyles

PLOS Genetics

On behalf of:
